# Structural basis for conductance through TRIC cation channels

Min Su[1,2,*], Feng Gao[1,2,*], Qi Yuan[3,*], Yang Mao[1,2,4,*], De-lin Li[5,*], Youzhong Guo[6], Cheng Yang[4], Xiao-hui Wang[5], Renato Bruni[7], Brian Kloss[7], Hong Zhao[1,2], Yang Zeng[5], Fa-ben Zhang[5], Andrew R. Marks[3], Wayne A. Hendrickson[3,6,7] & Yu-hang Chen[1,2,5]

Mammalian TRICs function as $K^+$-permeable cation channels that provide counter ions for $Ca^{2+}$ handling in intracellular stores. Here we describe the structures of two prokaryotic homologues, archaeal SaTRIC and bacterial CpTRIC, showing that TRIC channels are symmetrical trimers with transmembrane pores through each protomer. Each pore holds a string of water molecules centred at kinked helices in two inverted-repeat triple-helix bundles (THBs). The pores are locked in a closed state by a hydrogen bond network at the C terminus of the THBs, which is lost when the pores assume an open conformation. The transition between the open and close states seems to be mediated by cation binding to conserved residues along the three-fold axis. Electrophysiology and mutagenesis studies show that prokaryotic TRICs have similar functional properties to those of mammalian TRICs and implicate the three-fold axis in the allosteric regulation of the channel.

[1] State Key Laboratory of Molecular Developmental Biology, Institute of Genetics and Developmental Biology, Chinese Academy of Sciences, Beijing 100101, China. [2] CAS Center for Excellence in Biomacromolecules, Beijing 100101, China. [3] Department of Physiology and Cellular Biophysics, Columbia University, New York, New York 10032, USA. [4] State Key Laboratory of Medicinal Chemical Biology, College of Pharmacy and Tianjin Key Laboratory of Molecular Drug Research, Nankai University, Haihe Education Park, 38 Tongyan Road, Tianjin 300353, China. [5] University of Chinese Academy of Sciences, Beijing 100049, China. [6] Department of Biochemistry and Molecular Biophysics, Columbia University, New York, New York 10032, USA. [7] Center on Membrane Protein Production and Analysis, New York Structural Biology Center, 89 Convent Avenue, New York, New York 10027, USA. * These authors contributed equally to this work. Correspondence and requests for materials should be addressed to Y.-h.C. (email: yuhang.chen@genetics.ac.cn).

As an essential secondary messenger in cellular signal transduction pathways, calcium ions ($Ca^{2+}$) take part in important cellular functions including muscle contraction, cell growth and cell death[1–4]. The sarcoplasmic/endoplasmic reticulum (SR/ER) is a complex intracellular $Ca^{2+}$-storage system[5], from which ryanodine receptors (RyRs) and inositol 1,4,5-triphosphoate receptors (IP$_3$Rs) serve as channels to release the flow of $Ca^{2+}$ for muscle activation and other activities[6–13]. The efflux of $Ca^{2+}$ from the SR leads to the accumulation of a transient negative potential within the SR/ER lumen, which would be expected to inhibit the process of the subsequent release of $Ca^{2+}$ (refs 14–16). Therefore, robust counter-ion currents are vital to balance potential across the SR/ER membrane and to maintain efficient $Ca^{2+}$ release and re-uptake[15–19]. Counter currents, such as $K^+$, $Mg^{2+}$ or $Cl^-$ (refs 17,20–22), are anticipated but which ion channels are involved has been debated[23–25].

Miller and co-workers first identified and characterized $K^+$ channel activity from SR membranes[26,27]. Subsequently, Takeshima and co-workers cloned, characterized and named the trimeric intracellular cation channel TRIC-A, suggesting that it would be likely to act as a counter-ion channel to neutralize the transient luminal negative charge caused by $Ca^{2+}$ release from the intracellular stores[23]. Conductance properties observed for TRIC-A were similar to those of the previously identified SR $K^+$ channels, except for a lack of sensitivity to decamethonium[28]. The relationship between TRIC channels and other SR $K^+$ channels remains incompletely settled. However, an emerging concept is that TRIC subtypes form SR $K^+$ channels[29]. As evidenced by the fact that their genetic ablation led to compromised $K^+$ permeability and $Ca^{2+}$ release across the SR/ER membrane[30], TRICs play important roles as counter-ion channels for balancing the charge potential change during $Ca^{2+}$ release/uptake in the intracellular stores[31–33].

In mammals, TRIC channels have two isoforms with distinctive regulatory properties: TRIC-A is abundantly expressed in SR of excitable cells, while TRIC-B is expressed ubiquitously in ER of non-excitable cells[23,32,33]. Knockout mice lacking both TRIC-A and TRIC-B channels suffer from embryonic heart failure. Mutations in TRIC-A have been linked to hypertension and muscular diseases, whereas mutations in TRIC-B have been linked to bone and pulmonary diseases[5,31,34]. TRIC channels are selective for monovalent cations, with the permeability ratio being 1.5 for $K^+$ over $Na^+$ (ref. 23). TRIC channels are not homologous to any other channel characterized to date and hence may belong to a novel class of ion channels.

The mammalian TRIC proteins contain ∼300 amino acids and are characterized by a conserved N-terminal transmembrane (TM) domain and a diverse hydrophilic C-terminal tail of ∼60 amino acids. Initially, mammalian TRICs were predicted to contain three membrane-spanning segments[23]. However, a later analysis convincingly showed that these contained seven TM segments[35]. This analysis also brought the realization that TRIC homologues occur in other domains of life, including insects, algae, archaea and bacteria, identified as the UPF0126 family[35]. Interestingly, the prokaryotic homologues are shorter (∼210 residues), only containing the predicted TM domain of the eukaryotic counterpart. A systematic topological analysis indicated that the TRIC family might have arisen by duplication of an ancestral 3-TM segment followed by addition of a C-terminal segment to generate the eventual 7-TM protein[35]. To the best of our knowledge, no prokaryotic TRIC has been functionally characterized thus far, and detailed structural information is not available for any prokaryotic TRIC channel protein.

In this study, we analyse the structure and function of a bacterial and an archaeal TRIC channel. Both proteins have similar structures, consisting in symmetric trimers of 7-TM subunits, each composed as inverted quasi-repeats of triple-helix bundles (THBs) plus an added TM segment (TM$_7$). Each subunit possesses an apparent ion conduction pathway, which in the structure at the highest resolution (1.6 Å) is marked by a string of water molecules. In the close state, the C-terminal THB is locked by a network of hydrogen bonds. This network is not observed in an alternative conformation where the pore is open. The transition between these conformations is associated with the binding of cations to a conserved site on the three-fold axis.

## Results

**Sequence analysis of the TRIC-related proteins.** Bioinformatics analysis revealed that eukaryotic TRICs were mainly from animals, none from plants, and only a few from green algae. To better understand how TRIC proteins are represented in the animal kingdom, which includes various vertebrates and invertebrates, we clustered 425 non-redundant sequences into a superfamily of animal TRICs at the PSI-BLAST level $E \leq 10^{-3}$, then into two distinct families of vertebrate and invertebrate TRICs at an initial threshold of $E \leq 10^{-60}$, and finally into subfamilies of TRIC-A and TRIC-B (for the vertebrate TRICs) at a typical initial thresholds of $E \leq 10^{-90}$. We also analysed 100 non-redundant sequences from prokaryotes at the PSI-BLAST level $E \leq 10^{-4}$. Details are in Supplementary Table 1.

Sequences identified as prokaryotic TRICs show a 7-TM pattern as for the eukaryotic TRICs. Moreover, the conservation patterns are substantially similar for the two groups even though pairwise sequence identity levels are low. A structure-based alignment of the human TRIC-A and TRIC-B sequences with two prokaryotic TRICs, chosen because structures were forthcoming for these, is shown in Fig. 1a where patterns of sequence conservation calculated from the individual groups of prokaryotic and eukaryotic TRICs are also compared.

**Protein production and biochemical characterization.** Using a structural genomics approach[36], we tested expression for 53 bacterial and archaeal likely homologues, screened for detergent choice and stability on 8 of them, and obtained suitable crystals for 2 of them. The prokaryotic homologues from the archaeon *Sulfolobus acidocaldarius* (*Sa*TRIC) and the bacterium *Colwellia psychrerythraea* (*Cp*TRIC) were found to be trimeric both by size-exclusion coupled with multi-angle light-scattering measurement and by chemical crosslinking experiments (Fig. 1b). In our ultimate structure-based sequence alignment, *Sa*TRIC shares 18% and 23% sequence identity to *Hs*TRIC-A and *Hs*TRIC-B, respectively; while *Cp*TRIC shares 18% to *Hs*TRIC-A and 19% to *Hs*TRIC-B.

**Crystallization and structural determination.** The archaeal *Sa*TRIC protein was solubilized by various detergents and crystallized readily, but diffracted poorly. Improved diffraction was obtained after the application of lipid cubic phase (LCP). When solubilized in decylmaltoside (DM), *Sa*TRIC was crystallized in LCP in different lattices, including P6$_3$, P321 and R32, from various conditions[37]. All these different lattices contain one *Sa*TRIC molecule per asymmetric unit (ASU), but with distinctive packing and diffraction ability. Phases were determined initially at 3.1 Å resolution (type 1, in P321 lattice) by Se-single-wavelength anomalous diffraction (Se-SAD) from a mutant variant (L148M) of the selenomethionyl (Se-Met) protein. The structure for the wild-type protein in another lattice was solved by molecular replacement and extended to 1.6 Å resolution (type 2a, in P6$_3$ lattice; Fig. 1c). The final model for *Sa*TRIC was refined to $R_{work}/R_{free}$ values of 16.8%/17.9%, containing ordered

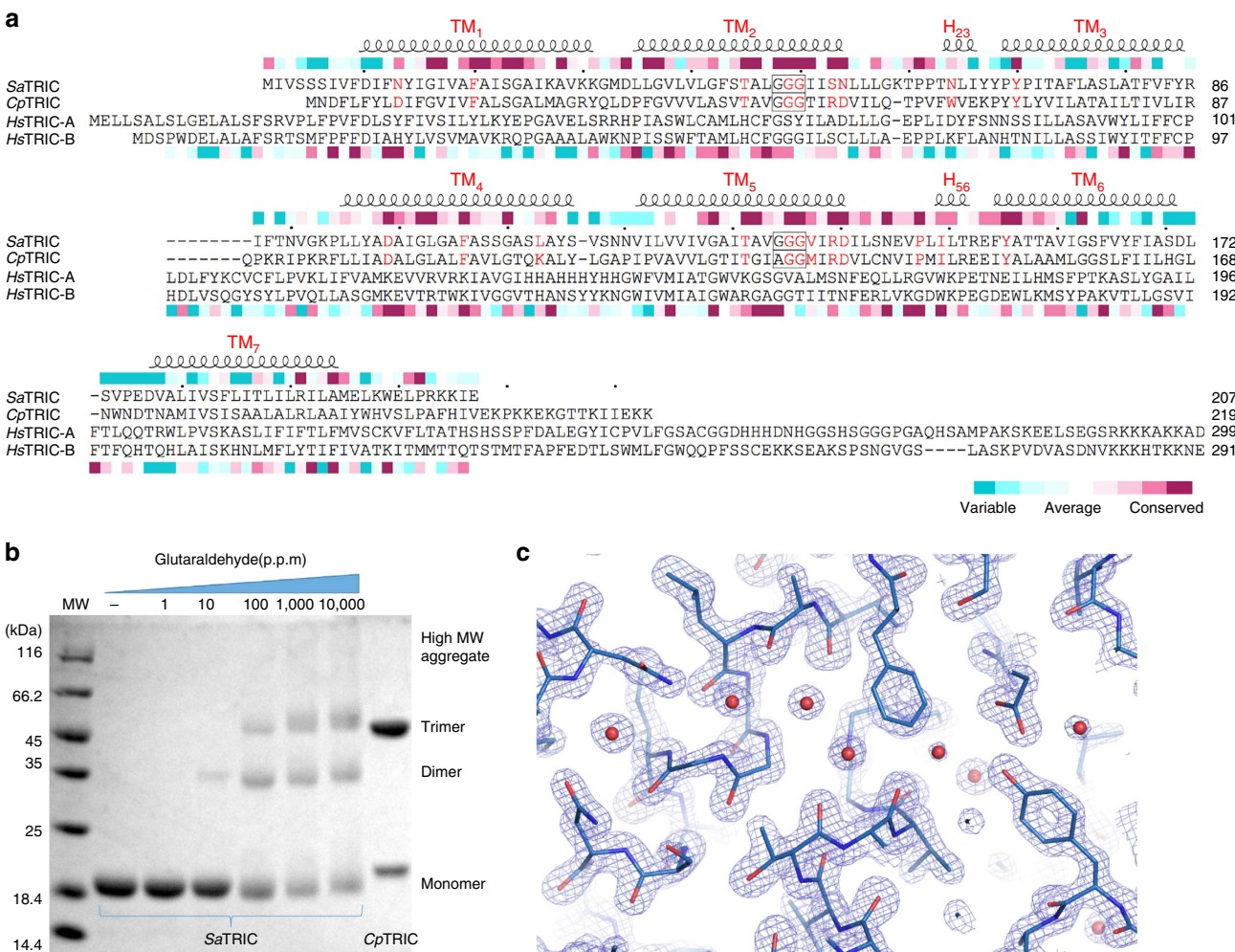

**Figure 1 | Sequence analysis and characterization for the TRIC proteins.** (**a**) Structure-based sequence alignment of prokaryotic TRICs from *Sa*TRIC, bacterial *Cp*TRIC, and human TRIC-A and TRIC-B. The structures of both *Sa*TRIC and *Cp*TRIC have been used to restrict sequence gaps to inter-helical segments. Superior coils define extents of the helical segments; boxes are drawn for the highly conserved GGG signature motifs; red letters mark residues in the prokaryotic TRIC that are involved in the ion conduction pathway; and the coloured inferior bar encodes ConSurf sequence variability[54] for the prokaryotic TRIC of 100 non-redundant proteins (top) and eukaryotic TRIC of 169 non-redundant proteins (bottom). (**b**) Crosslinking of purified *Sa*TRIC and *Cp*TRIC in detergent micelles. The purified proteins were incubated with increasing amounts of glutaraldehyde, and samples were analysed by SDS–PAGE. *Sa*TRIC showed gradually increasing crosslinked trimeric species with increasing glutaraldehyde, whereas *Cp*TRIC mainly maintained a trimeric state, even in the absence of glutaraldehyde and the presence of SDS. MW, molecular weight. (**c**) Electron density map of *Sa*TRIC at 1.6 Å resolution. The initial phases were determined at 3.1 Å by Se-SAD, and further extended to 1.6 Å into the type 2a crystal (with a space group of P6₃) by molecular replacement. A section of the experimental map is superimposed onto the *Sa*TRIC model as refined at 1.6 Å resolution, contours are at 2.0σ.

residues 7–200, two $Na^+$ ions and 90 water molecules. Different from *Sa*TRIC, the bacterial *Cp*TRIC was solubilized in octylmaltoside (OM) and crystallized from detergent micelles in space group R32, with one protomer per ASU. This structure was solved to 2.4 Å resolution by Se-SAD. The final model for *Cp*TRIC was refined to $R_{work}/R_{free}$ values of 24.0%/27.5%, containing ordered residues 3–198, 5 $Cd^{2+}$ ions and 27 water molecules. To further identify ion binding or to capture alternative conformation of the channel structures, extensive efforts have been pursued under various conditions. Details for crystallization, data collection, structure refinement and analyses are shown in Table 1 and Supplementary Table 2. In the following, if not specified, we will focus on analyses of the high-resolution *Sa*TRIC structure (type 2a).

**Structural analyses on TRIC channels.** Both *Sa*TRIC and *Cp*TRIC, as crystallized in various lattices, are highly similar

symmetrical trimers (Fig. 2a), consistent with the initial characterization of TRIC-A from rabbit skeletal muscle[23] as a trimer. Those subunits are tightly associated, burying ~5,500 Å² (*Sa*TRIC) of total surface area within trimer interfaces. The electrostatic potential surface is largely negative on the extracellular surface (Fig. 2b) and largely positive on the cytoplasmic surface (Fig. 2c). Each protomer of prokaryotic TRIC comprises seven TM helices (termed TM₁–TM₇; Fig. 2d–f), as predicted[35]. However, to the best of our knowledge, the folding is novel. The membrane orientation is specified experimentally from green fluorescent protein tagging of *Escherichia coli* YadS (prokaryotic TRIC homologue in *E. coli*), which is consistent with the positive-inside rule as applied to the electrostatic potential surfaces of the structure[38].

Overall, the two prokaryotic TRICs are very similar, with a root mean squared deviation (r.m.s.d.) of 1.39 Å when 188 Cα atoms are superimposed. All of the TM helices are fully aligned, except for some connecting loops (Fig. 2g,h). The TRIC sequences

**Table 1 | Data collection and refinement statistics.**

| Protein name | SaTRIC | | | | CpTRIC |
|---|---|---|---|---|---|
| Crystal name | Type 1 | Type 2a | Type 2b | Type 3 | Se-Met |
| Wavelength (Å) | 0.97853 | 0.97876 | 0.8153 | 0.91532 | 0.97853 |
| Resolution range (Å) | 42.8-3.10 (3.31-3.10) | 32.1-1.60 (1.66-1.60) | 46.0-1.90 (1.97-1.90) | 35.5-2.40 (2.49-2.40) | 33.5-2.40 (2.49-2.40) |
| Space group | P321 | P $6_3$ | P $6_3$ | R 3 2 | R 3 2 |
| Unit cell: a, b, c (Å) | 111.1 111.1 47.7 | 64.3 64.3 80.8 | 64.6 64.6 81.0 | 115.9 115.9 134.8 | 91.3 91.3 252.8 |
| Total reflections | 164,071 | 795,280 (35,707) | 307,291 (30,734) | 110,982 (6,776) | 1197,373 (64,357) |
| Unique reflections | 6,373 (1,127) | 24,834 (2,374) | 15,160 (1,502) | 12,201 (1,126) | 16,250 (1,588) |
| Multiplicity | 25.7 (26.8) | 32.0 (15.0) | 20.3 (20.5) | 9.1 (6.0) | 73.7 (40.5) |
| Completeness (%) | 1.00 (1.00) | 0.99 (0.95) | 1.00 (1.00) | 0.88 (0.83) | 1.00 (1.00) |
| Mean $I$/sigma($I$) | 21.0 (1.0) | 29.5 (1.1) | 18.2 (1.6) | 14.0 (0.9) | 24.9 (1.1) |
| Wilson B-factor | 106.4 | 32.3 | 22.5 | 55.3 | 61.8 |
| R-merge | 0.128 (4.6) | 0.092 (2.4) | 0.178 (2.2) | 0.114 (1.7) | 0.137 (4.1) |
| R-meas | 0.131 (4.7) | 0.094 (2.4) | 0.183 (2.3) | 0.121 (1.9) | 0.138(4.2) |
| R-pim | 0.026 (0.88) | 0.016 (0.611) | 0.040 (0.50) | 0.039 (0.73) | 0.016 (0.65) |
| CC1/2 | | 1 (0.284) | 0.999 (0.621) | 0.999 (0.404) | 1 (0.584) |
| CC* | | 1 (0.666) | 1 (0.876) | 1 (0.758) | 1 (0.859) |
| Reflections used in refinement | | 24,832 (2,373) | 15,144 (1,502) | 12,194 (1,124) | 16,232 (1,579) |
| Reflections used for R-free | | 1,248 (125) | 759 (75) | 595 (54) | 811 (81) |
| R-work | | 0.168 (0.278) | 0.187 (0.244) | 0.225 (0.326) | 0.240 (0.379) |
| R-free | | 0.179 (0.314) | 0.220 (0.274) | 0.253 (0.362) | 0.275 (0.445) |
| CC (work) | | 0.883 (0.456) | 0.933 (0.834) | 0.938 (0.636) | 0.803 (0.725) |
| CC (free) | | 0.931 (0.361) | 0.926 (0.779) | 0.959 (0.623) | 0.783 (0.604) |
| Number of non-hydrogen atoms | | 1,566 | 1,632 | 1,530 | 1,487 |
| Macromolecules | | 1,474 | 1,517 | 1,517 | 1,456 |
| Ligands | | 2 | 1 | 5 | 4 |
| Solvent | | 90 | 114 | 8 | 27 |
| Protein residues | | 194 | 197 | 203 | 196 |
| R.m.s. (bonds) | | 0.003 | 0.005 | 0.008 | 0.002 |
| R.m.s. (angles) | | 0.51 | 0.6 | 0.97 | 0.42 |
| Ramachandran favoured (%) | | 98 | 98 | 96 | 99 |
| Ramachandran allowed (%) | | 2.1 | 2.5 | 4.5 | 1 |
| Ramachandran outliers (%) | | 0 | 0 | 0 | 0 |
| Rotamer outliers (%) | | 0 | 0.61 | 0.62 | 0 |
| Clashscore | | 1.65 | 4.16 | 7.35 | 2.66 |
| Average B-factor | | 26.13 | 24.57 | 59.08 | 75.03 |
| Macromolecules | | 25.12 | 23.4 | 59 | 74.97 |
| Ligands | | 25.55 | 21.5 | 79.72 | 121.64 |
| Solvent | | 42.67 | 40.23 | 60.85 | 71.25 |
| PDB ID | | 5WUC | 5WUD | 5WUE | 5WUF |

*Values in parentheses are for highest-resolution shell.

contain approximate internal repeats, with sequence identities of ~31% and ~28% for the repeat pairs in SaTRIC and CpTRIC, respectively (Supplementary Fig. 1a). Interestingly, each homologous repeat forms a structurally similar THB, with r.m.s.d. of 1.32 Å/79 $C_\alpha$ for SaTRIC and r.m.s.d. of 1.45 Å/71 $C_\alpha$ for CpTRIC. More remarkably, the N-THB (TM$_{1-3}$) and C-THB (TM$_{4-6}$) assemble into a novel structural fold, with a quasi-two-fold symmetry axis parallel to the membrane plane (Supplementary Fig. 1b). Both N-THB and C-THB include short juxtamembrane helices, H$_{2,3}$ between TM$_2$ and TM$_3$ on the extracellular side and H$_{5,6}$ between TM$_5$ and TM$_6$ on the intracellular side. Unique helix TM$_7$ is located on the periphery of the trimer, making contact with TM$_4$ and TM$_6$ from the C-THB; it is poorly conserved except at its cytoplasmic C-terminal end.

**Structure of the ion conduction pathway**. Many other trimeric channels, including ATP-gated P2X receptors[39], acid-sensing ion channels[40] and the mechanosensitive Piezo1 channels[41], form central ion-conducting pores at the three-fold axes of protomer association; by contrast, the prokaryotic TRIC trimers have three pores as calculated by the HOLE programme[42], one through each protomer (Fig. 3a). In the SaTRIC structure at 1.6 Å resolution, an intriguing set of nine electron density features occupies each pore (Fig. 3b). We identified these densities as water molecules on the basis of their coordination with the surrounding residues.

The TRIC channel pore is bounded by four helices, two from each THB (TM$_{1-2}$ from N-THB and TM$_{4-5}$ from C-THB) as associated antiparallel about an axis of quasi-diad symmetry (Fig. 3c). Each pore has an opening to the extracellular side, and it tapers gradually from outside to two narrow constrictions in the middle and finally to a seal at the cytosolic side. Overall, the pore is lined with highly conserved and generally hydrophobic residues, as indicated from the analysis of sequence conservation of prokaryotic TRIC proteins (Fig. 3d). Despite this hydrophobicity, the electrostatic potential on the pore surface is polarized (Fig. 3e), presumably due to sharp invagination adjacent to charged residues outside the membrane. The generally electronegative character of the pore surface may be consistent with its function in conducting cations across the membrane.

The opening of the pore at the extracellular surface involves N-THB residues at one end of that THB; and, by virtue of the

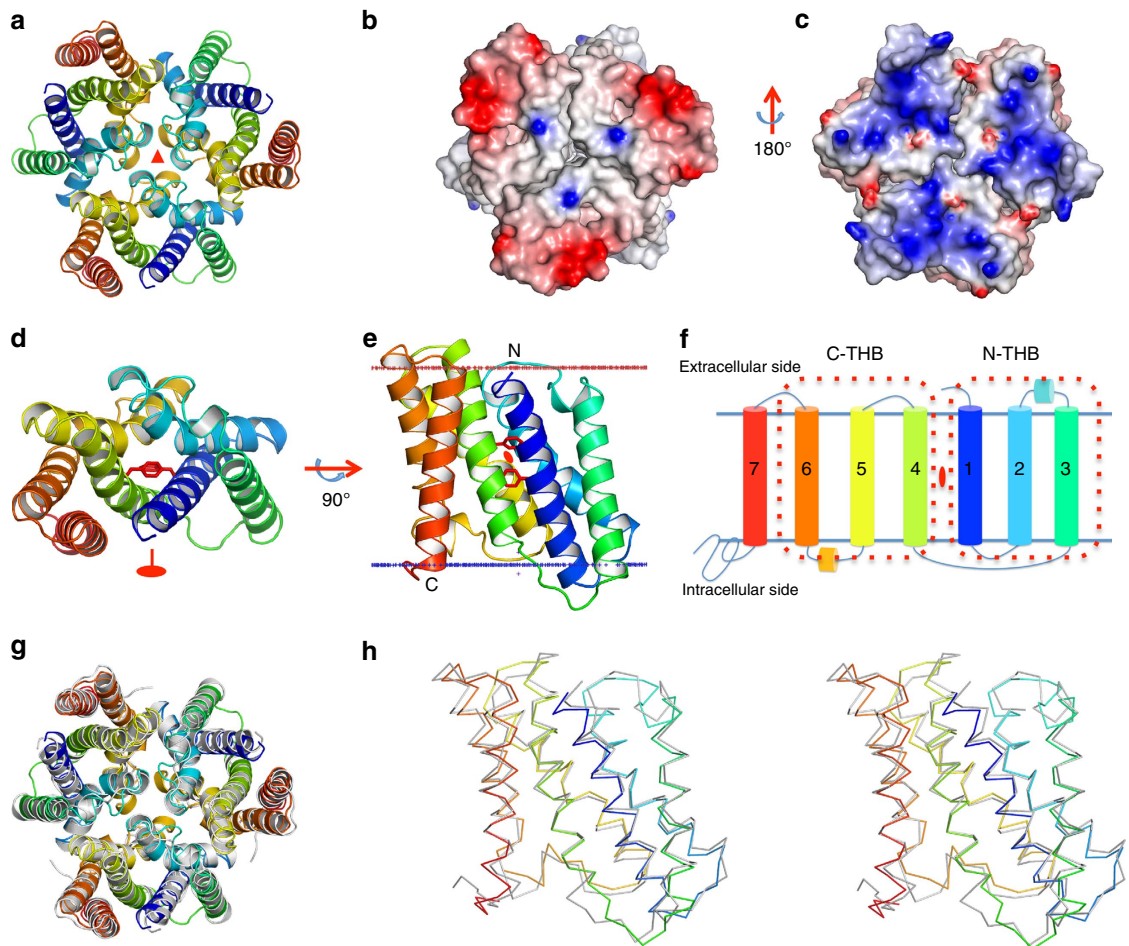

**Figure 2 | Crystal structure of prokaryotic TRICs.** (**a**) Ribbon drawing of the *Sa*TRIC trimer, as viewed from outside of the membrane. The colouring is spectral for each protomer, from dark blue at its N terminus to red at its C terminus. (**b,c**) Electrostatic potential at the solvent accessible contact surfaces[55] as viewed from the extracellular side (**b**), as in **a**, and from the intracellular side (**c**), 180° from **b**. The contour level is at $\pm 7\,\mathrm{kTe}^{-1}$. Red is for negative potential and blue is for positive potential. (**d,e**) Ribbon drawing of the *Sa*TRIC protomer, coloured as in **a**. Side chains of conserved residues F20 and F106 are shown as red stick. Top view (**d**), as viewed from outside of the membrane; front view (**e**), looking towards the three-fold axis from within the lipid bilayer, 90° rotation from **d**. Membrane boundaries were calculated by the Orientations of Proteins in Membranes (OPM) server. (**f**) Membrane topology diagram for prokaryotic TRICs; $\mathrm{TM}_{1\text{-}3}$ constitutes N-THB and $\mathrm{TM}_{4\text{-}6}$ constitutes C-THB. Spectral colouring is as in **e**. (**g**) Superimposition of the trimeric structure of *Sa*TRIC (type 2a) and *Cp*TRIC (Se-Met). *Sa*TRIC protomers are coloured as in **a**, and *Cp*TRIC protomers are coloured in grey. (**h**) Superimposition of the protomer structure of *Sa*TRIC (type 2a) and *Cp*TRIC (Se-Met). Stereo view of the superimposed $C_{\alpha}$ backbones, viewed as in **e** and coloured as in **g**.

quasi-diad symmetry, the blockage of the pore at the cytoplasmic surface involves equivalent residues from the C-THB. The cytoplasmic closure of the pore is due to a hydrogen-bonded network of interactions among C-THB residues D99 ($\mathrm{TM}_4$)–R139-D140 ($\mathrm{TM}_5$)–Y155 ($\mathrm{TM}_6$) (Fig. 3f). We describe this arrangement as a 'locked' conformation of C-THB. By contrast, interactions among the quasi-symmetric mates in N-THB, N13 ($\mathrm{TM}_1$)–S53-N54 ($\mathrm{TM}_2$)–Y70 ($\mathrm{TM}_3$), are mostly undone; thereby, N-THB is released to an unlocked, pore-open conformation (Fig. 3g). This asymmetry in the inverted pair of THBs, with C-THB in a locked state versus N-THB in an unlocked state, generates an open-outward conformation. When the two THBs are superimposed (Fig. 3h), it is evident that the network C-THB H-bond interactions of D99 ($\mathrm{TM}_4$) with R139 ($\mathrm{TM}_5$) and Y155 ($\mathrm{TM}_6$) at the cytoplasmic side, together with the side chain of L148 from helix $\mathrm{H}_{5,6}$, serve to close the ion translocation passageway. Moreover, $\mathrm{TM}_4$ in C-THB is slightly kinked in its middle, by $\sim 16°$, whereas $\mathrm{TM}_1$ in N-THB is straight; thereby, the pore is relatively open at the unlocked extracellular surface. We will see below that conformational changes within the C-THB can

switch it to unlocked state, as would be needed for activation of the channel. Residues of the H-bonded network are highly conserved in prokaryotes but not in eukaryotic TRICs (Fig. 1a).

A signature motif in prokaryotic TRIC sequences is a highly conserved stretch of glycine residues found repeated within $\mathrm{TM}_2$ and $\mathrm{TM}_5$ (Fig. 1a). The eukaryotic TRICs show similar features, though not as glycine rich (Fig. 1a). For *Sa*TRIC, these stretches are GGG48–50 in $\mathrm{TM}_2$ and GGG134–136 in $\mathrm{TM}_5$. The high-resolution structure clearly reveals that these glycine motifs generate similar kinks ($\sim 41°$ in $\mathrm{TM}_2$ and $\sim 45°$ in $\mathrm{TM}_5$) near the middle of the respective TM segments (Fig. 4a,b and Supplementary Fig. 1c), disrupting the normal $\alpha$-helical hydrogen-bonding pattern and exposing backbone amines that coordinate bridging water molecules (Fig. 4c). These intervening water molecules are from the centre of the string of pore waters (Fig. 3b), and these positions correspond to two narrow constrictions in the middle of the putative ion conduction passageway where the pore is covered by the phenyl groups of opposing residues: F20 ($\mathrm{TM}_1$) and F106 ($\mathrm{TM}_4$) (Fig. 4d,e). It is tempting to speculate that these glycine-mediated kinks may

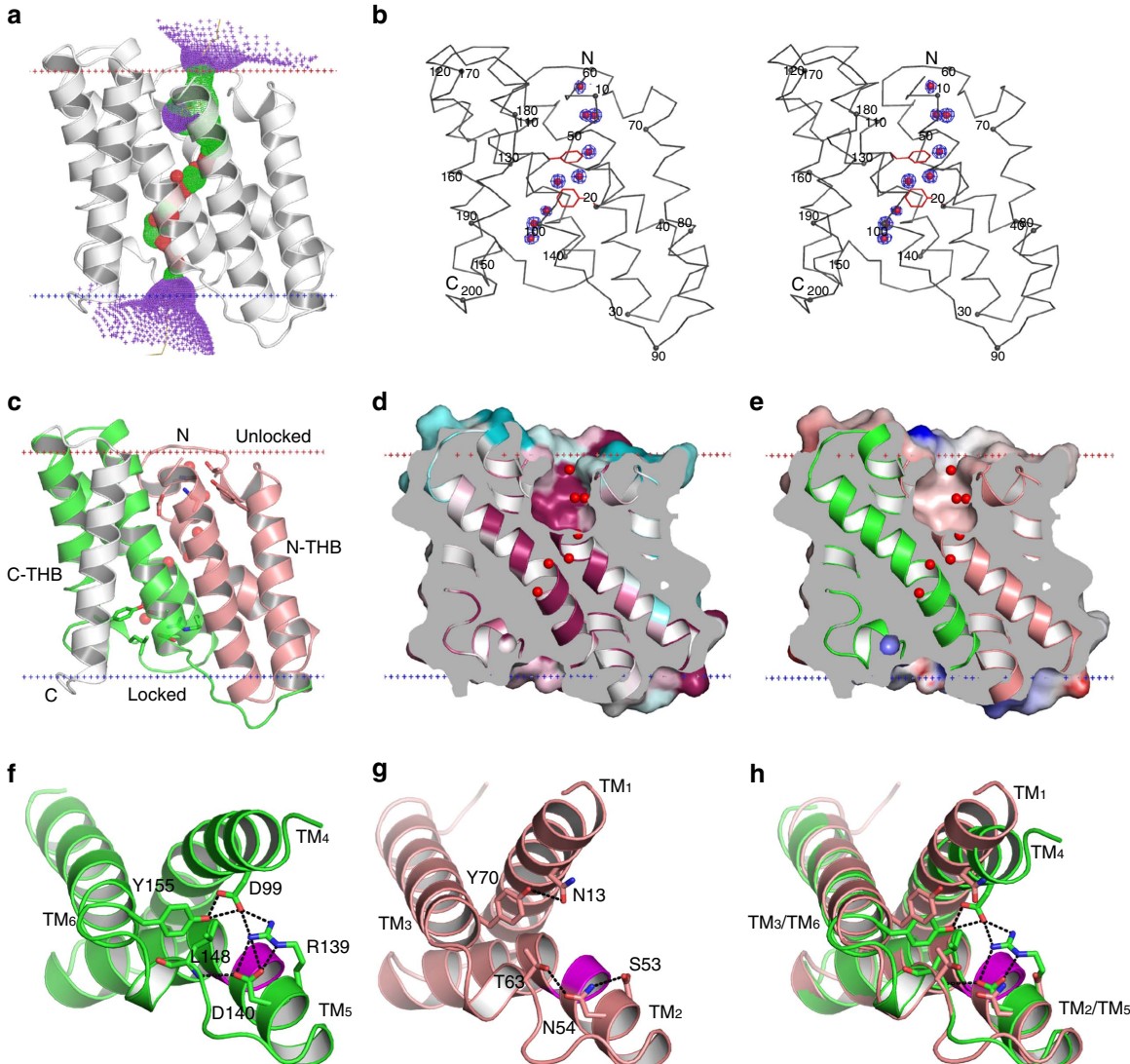

**Figure 3 | The ion conduction pathway of prokaryotic TRIC.** (**a**) The pore-lining surface as computed by the programme HOLE is drawn into a ribbon model of the *Sa*TRIC structure (type 2a). We used a simple van der Waals surface for the protein and the programme default probe radius of 1.15 Å. The pore at radius below 1.15 Å is shown in red and that above 2.3 Å is shown in purple, and the intermediate zone is in green. A yellow line through the channel marks the calculated centre line of the pore. (**b**) Stereo view of the $C_\alpha$ backbone of *Sa*TRIC, oriented as in Fig. 2e. Two conserved residues, F20 (TM$_1$) and F106 (TM$_4$), were shown in red stick. Water molecules observed within the ion conduction pathway are shown as red sphere. Density contours are shown for the water molecules. (**c**) Ribbon drawing of SaTRIC protomer as in **a**, but with N-THB (TM$_{1-3}$) in salmon, C-THB (TM$_{4-6}$) in green and TM$_7$ in grey. The observed water molecules are shown as **b**. (**d**,**e**) Cross-section through the *Sa*TRIC (type 2a). The models are viewed as **c**; surface conservation is shown in **d** and electrostatic potential is shown in **e**. (**f**) Ribbon drawing of C-THB in the locked state of *Sa*TRIC type 2a. The inter-helices network of D99–R139–D140–Y155 is indicated. The GGG motif (TM$_5$) is shown in magenta. (**g**) Ribbon drawing of N-THB, in its unlocked state and oriented as for the C-THB. The corresponding symmetry mates are shown for comparison. The GGG motif (TM$_2$) is shown in magenta. (**h**) The superimposed N-THB and C-THB, coloured as in **f**,**g**. All membrane boundaries were calculated as Fig. 2e.

contribute to K$^+$ selectivity. However, extensive attempts to identify potential cation-binding site within the channel pore with crystals grown in various conditions, containing K$^+$, or Rb$^+$, or Cs$^+$, were unsuccessful so far (for details, see Supplementary Table 2).

**Functional characterization.** Although mammalian TRICs are known as K$^+$ channels[23,29,30,33], the function of prokaryotic TRIC had not been previously characterized. To test its function, purified *Sa*TRIC protein was fused into planar lipid bilayers with 210 mM KCl in both *trans* and *cis* sides (details in Methods). Application of a series of TM potentials to the *trans* side resulted

in apparent unitary currents with a linear single-channel *I–V* relationship (Fig. 5a,b). An asymmetrical solution (210 mM KCl in *cis*; 810 mM KCl in *trans*) was applied to confirm that *Sa*TRIC is permeable to K$^+$, showing a reversal potential (near −40 mV) similar to that predicted by the Nernst equation for K$^+$ permeability. The conductance level is $159 \pm 2$ pS for *Sa*TRIC, comparable to the $199 \pm 2$ pS for mouse TRIC-B[33]. To further investigate ion selectivity for *Sa*TRIC, reversal potentials were measured under various bi-ionic conditions. The relative cation permeability sequence for *Sa*TRIC is in the order: K$^+$ > Na$^+$ > Rb$^+$ > Mg$^{2+}$ ~ Ca$^{2+}$ (Fig. 5c and Supplementary Fig. 2a–d), similar to that of mouse TRIC-B[32]. Although the prokaryotic TRICs reveal three separated pores within the

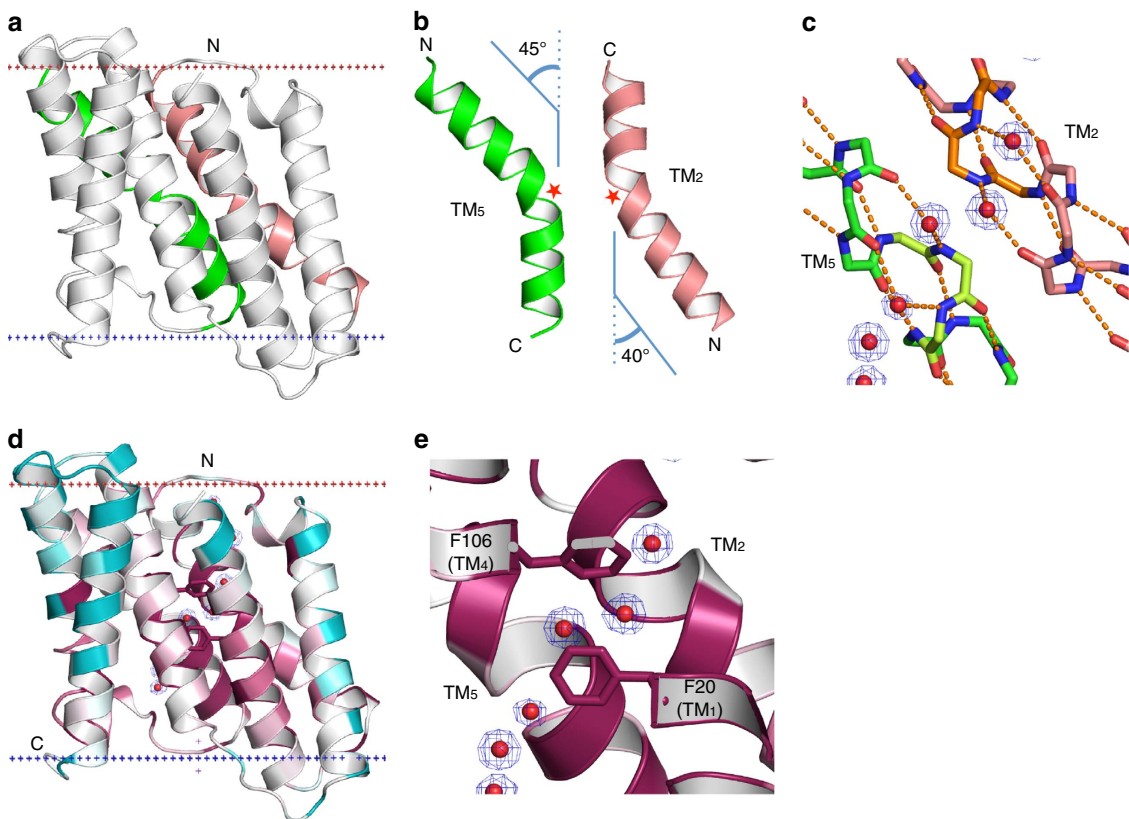

**Figure 4 | Characteristics of the ion conduction pathway.** (**a**) Ribbon diagram of SaTRIC protomer, with the kinked TM$_2$ (in salmon) and TM$_5$ (in green) helices, oriented as in Fig. 2e. (**b**) The kinked TM$_2$ (in salmon) and TM$_5$ (in green), with red stars marking the GGG motifs. (**c**) Hydrogen-bonding patterns for the kinked helices and associated water molecules from the ion conduction pathway. The GGG motif of TM$_2$ is in orange and GGG motif of TM$_5$ is in light green. Density contours are shown for the water molecules. (**d**) Ribbon diagram of SaTRIC protomer, viewed as in **a** and coloured by sequence conservation as in Fig. 1b. Two conserved residues, F20 (TM$_1$) and F106 (TM$_4$), are shown as stick. (**e**) A zoom view of the GGG motifs in the kinked TM$_2$ and TM$_5$, coloured as in **d**. Backbone-coordinated water molecules are shown as in **c**.

trimeric channel, they are not gated in a concerted fashion. Instead, there could be alternative conformers for each independent protomer, perhaps modulated by an allosteric effector. Indeed, we found that the SaTRIC channel gating transits between four predominant sub-conductance levels that were ~79, 66, 21 and 11% of the full open state (Fig. 5d), roughly similar to mouse TRIC-B[33,43]. We conclude that the prokaryotic SaTRIC also function as a K$^+$ channel, with similar electrophysiological properties as the eukaryotic TRIC channels.

On the basis of the structure of SaTRIC, we predicted five residues critical to channel gating (D99, R139, D140, L148 and Y155; Fig. 5g), which form a complex interaction network among helices in C-THB to lock the ion-translocation pore in a closed state. In symmetric bilayer experiments, SaTRIC R139A and L148A behaved similar to the wild type at 20 mV holding potential, whereas SaTRIC N144A and Y155A open more frequently at the same holding potential (Fig. 5e,h,i). Surprisingly, when tested in asymmetric bilayer experiments (Fig. 5f), SaTRIC D99A showed current oppositely directed to that of wild type. Bilayers incorporating SaTRIC D99A proved relatively unstable, but we were able to acquire current–voltage data showing reversal potential (Supplementary Fig. 2e,f). This apparent reversal of cation/anion selectivity is intriguing, and it suggests a key role of the pore passageway in ion permeation across the membrane.

**Channel gating.** The well-defined series of water molecules through each protomer of the high-resolution type 2a SaTRIC structure (Figs 3 and 4) and sensitivity of conduction properties to mutations of associated residues (Fig. 5) define a putative ion conduction pathway. This path for ion translocation is blocked; however, with C-THB in a locked state, the channel is in an open-outward but closed conformation (Fig. 6a). It is unclear what physiological stimuli could activate the channel and how to stabilize the channel in an open conformation, but a fortunate structure in another lattice (SaTRIC type 3, R32; Supplementary Table 2) may shed light on channel gating. This structure captures the pore open across the entire membrane (Fig. 6b), and the open pathway overlays well with the possible ion conduction path defined by water molecules in the 1.6 Å resolution structure (type 2a). In addition, some clear but unmodelled density was also observed within the same pathway (Supplementary Fig. 3).

The pore is more obvious and wider than that observed in the closed channel (type 2a), with a diameter of generally ~4 Å across the membrane. The overall structure is similar to that of the closed channel (type 2a), but there are several noticeable structural changes, including an unlocked conformation in C-THB and a newly formed connecting helix between N-THB and C-THB (Fig. 6c–e and Supplementary Fig. 4a,b). When either N-THB or C-THB is compared between the type 2a and type 3 structures, it is obvious to find that both of the GGG-motif helices, TM$_2$ and TM$_5$, are kinked to a larger degree (18° for TM$_2$, from 41° to 59°; and 7° for TM$_5$, from 45° to 52°; Fig. 6d). Furthermore, the C-THB converts from a locked state (type 2a) to an unlocked state (type3; Fig. 6e). Consistent with this observation, larger structural deviation occurs in C-THB versus N-THB, with r.m.s.d. of 1.67 Å for 79 C$_\alpha$ positions for the superimposed C-THB versus r.m.s.d. of 1.24 Å for 78 C$_\alpha$ positions for the superimposed N-THB (Supplementary Fig. 4c,d).

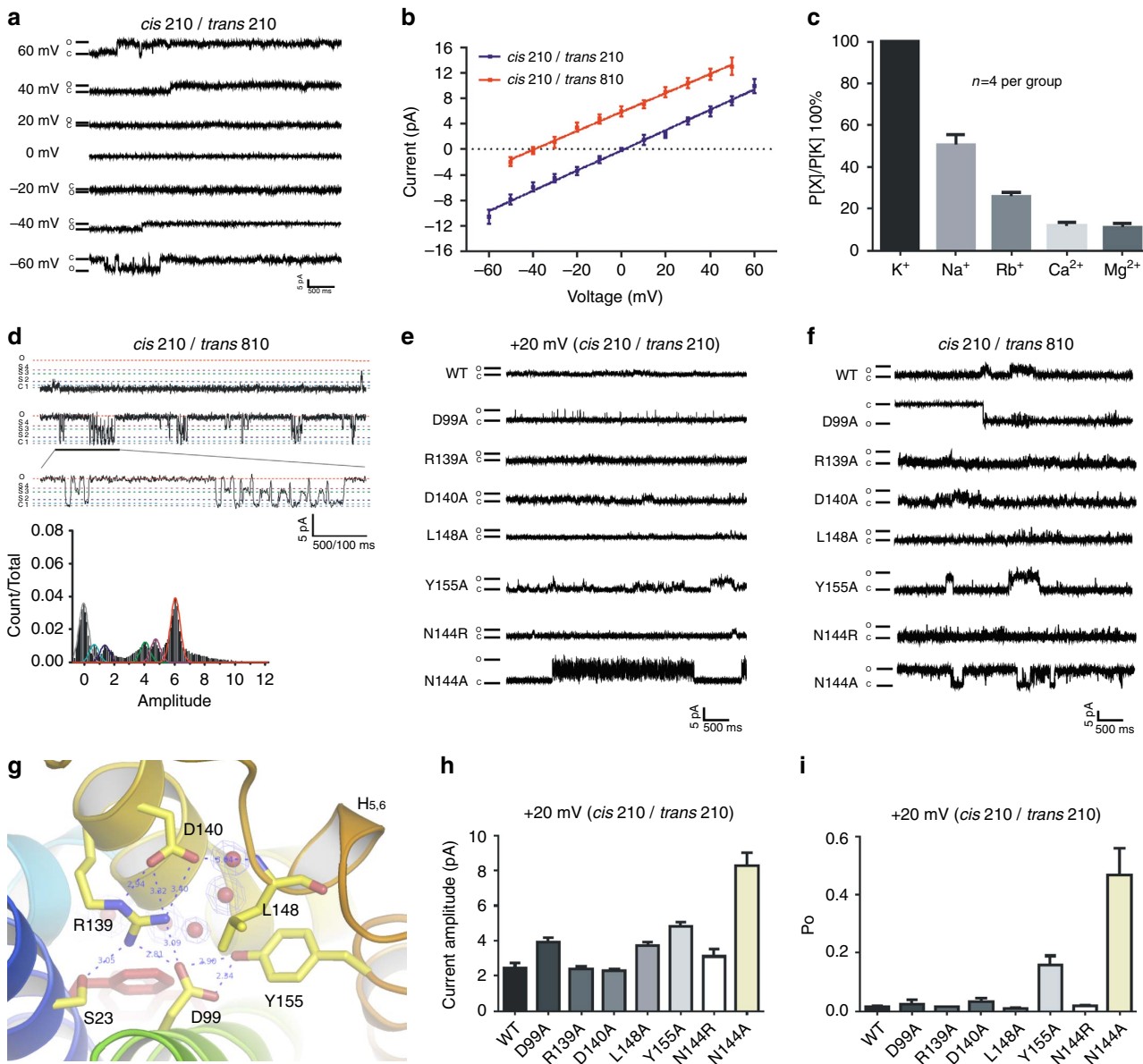

**Figure 5 | Ionic conductance measurements of the prokaryotic TRIC.** (**a**) Representative traces of single *Sa*TRIC currents recorded from planar lipid bilayers at different voltages (210 mM KCl in both *trans* and *cis* solutions). (**b**) Single *Sa*TRIC channel current–voltage relationship. Data are presented as mean ± s.e.m. (*n* = 6 for each point). (**c**) Relative cation permeability for *Sa*TRIC, *n* = 4 for each point. (**d**) Sub-conductance levels of the *Sa*TRIC, as in asymmetrical solutions, but here with 210 mM KCl in *cis* chamber and 810 mM in *trans* chamber. (**e**) Representative current traces of single *Sa*TRIC channels at + 20 mV (210 mM KCl in both *trans* and *cis* solutions) from wild-type and mutant proteins. (**f**) Representative current traces of single *Sa*TRIC channel at the asymmetrical solutions (210 mM KCl in *cis* and 810 mM KCl in *trans* chambers) from wild-type and mutant proteins. Interestingly, *Sa*TRIC D99A mutant displays a downward current, whereas wild type and other mutants show an upward current, suggesting *Sa*TRIC D99A is permeable to Cl⁻. (**g**) The intra-facial gate of the C-THB as locked by a complex H-bond interaction network involving residues tested by mutation. (**h**) Current amplitude for the wild-type and mutants of single *Sa*TRIC channel at + 20 mV holding potentials, *n* = 4 for each group. (**i**) Open probabilities for the wild-type and mutants of single *Sa*TRIC channel at + 20 mV holding potentials (same condition as in **e**).

Furthermore, with both unlocked, N-THB and C-THB become more similar with the r.m.s.d. after superpositions reduced to 1.17 Å/76 $C_\alpha$ (type 3) from 1.32 Å/79 $C_\alpha$ (type 2a) (Supplementary Fig. 4e,f). Altogether, the comparison of *Sa*TRIC in different conformations reveals a potential switch for opening of TRIC channels.

**Ion binding and channel modulation.** A highly conserved asparagine residue, near the C-terminal end of GGG-containing $TM_5$, is positioned near the three-fold axis, and $Na^+$ and $Mg^{2+}$ cations were found to be bound to this site (N144 in *Sa*TRIC) in

*Sa*TRIC structures (type2a and b; Supplementary Table 2 and Fig. 7a,b). Two monovalent $Na^+$ ions, each with three bound water molecules, were bound in the 1.6 Å-resolution type 2a structure (Fig. 7c), while in the 1.9 Å-resolution type 2b structure, obtained from a different condition, a single divalent $Mg^{2+}$ ion was bound at the same site, with slight adjustment on the N144 side chain and also with three bound water molecules (Fig. 7d). Since other trimeric ion channels (for example, acid-sensing ion channel, P2X receptor and Piezo-1) have ion-conducting pores along central three-fold axis, we initially considered this possibility for TRIC channels as well. Thus, we tested the *Sa*TRIC

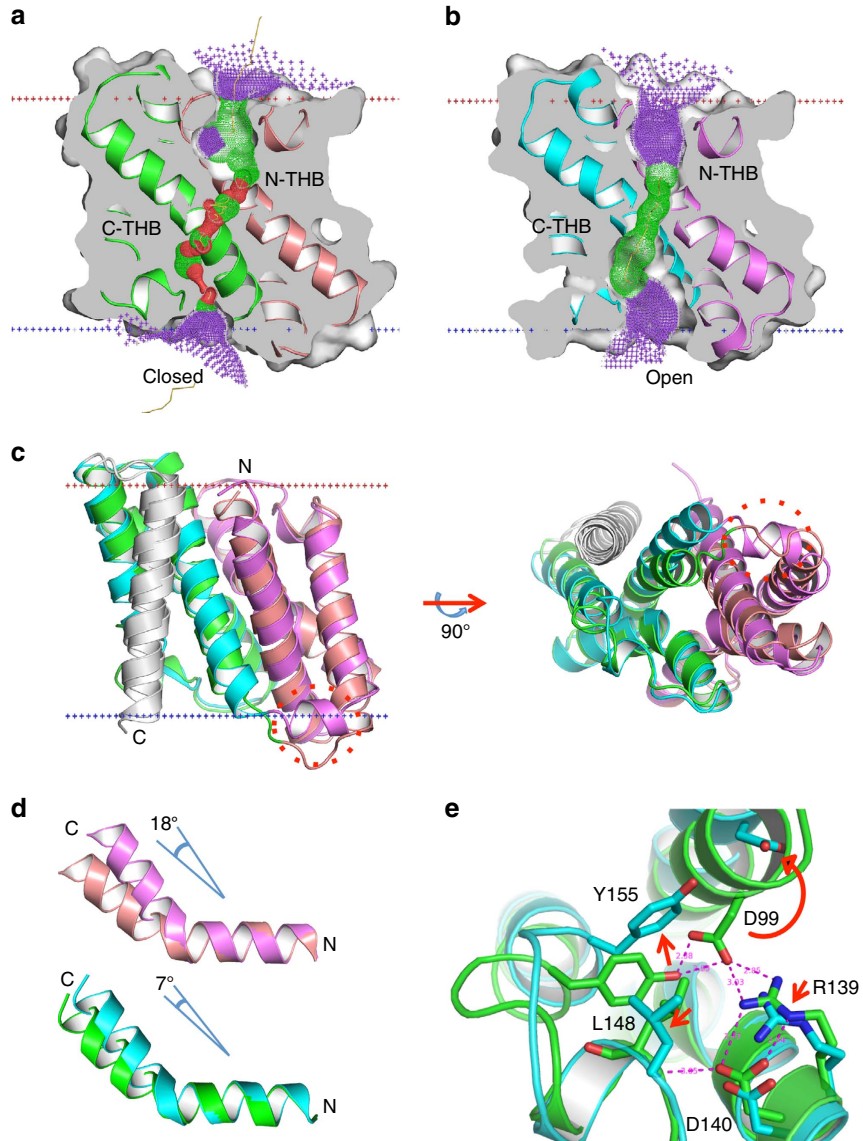

**Figure 6 | Gating of prokaryotic TRIC.** (**a**) Cross-section through the *Sa*TRIC in closed state (type 2a); N-THB in salmon, C-THB in green and TM$_7$ in grey. The superimposed pore drawing is coloured as in Fig. 3a. (**b**) Cross-section through the *Sa*TRIC in an open state (type 3); N-THB in violet, C-THB in cyan and TM$_7$ in grey. The superimposed pore drawing is coloured as in Fig. 3a. (**c**) The superimposition of *Sa*TRIC structures in different states: closed state (type 2a) versus open state (type 3), coloured as in **a** and **b**. (**d**) The superimposition of closed- and open-state structures at the GGG-kinked helices. TM$_2$ above and TM$_5$ below. Colouring is as in **a,b**. (**e**) Zoom-up view for the superimposed C-THBs: locked state (type 2a) versus unlocked state (type 3), coloured as in **a** and **b**. The surrounding residues of the intra-facial gate are shown.

N144R mutant, thinking that it might repel cations; however, it retained a similar conductance and open probability as the wild type, indicating that N144 is not directly involved in ion permeation.

Surprisingly, we found that ion binding to N144 is not present in the open-pore *Sa*TRIC structure (type 3). We presume that this change is because of the crystallization conditions (only Li$^+$ and no Na$^+$ or Mg$^{2+}$; Supplementary Table 2). The local environment around N144 then becomes more compact. This is evident from the distance between main-chain oxygen atoms of N144 about the three-fold axis (144O–144′O), which are slightly smaller in the ion-free state (6.2 Å, in type 3) versus the Na$^+$-bound state (7.1 Å, in type 2a) and also from distances associated with side-chain atoms of N144: ND$_2$–O′ 3.7→2.7 Å; OD$_1$–O′ 3.6→3.0 Å; and OD$_1$–OD$_1$′ 3.1→2.7 Å (Fig. 7e,f). It is not fully clear how these changes at the three-fold axis are coupled to unlocking of C-THB and opening of the *Sa*TRIC channel, but it

does seem that this is a site of channel modulation. In keeping with such allosteric regulation, single-channel recordings from the N144A mutant show a greatly increased open probability and larger conductance (Fig. 5h,i).

Another ion-binding site was also observed at an inter-protomer interface in the trimeric channel of *Cp*TRIC (Supplementary Fig. 5). One Cd$^{2+}$ ion was bound to a well-conserved Asp residues (D40 in *Cp*TRIC, located at the N-terminal end of the GGG-kinked TM$_2$), and interacting with several backbone carbonyl oxygen or nitrogen from the short linker between TM$_5$′ and TM$_6$′ on the adjacent protomer (residues 145–149 in *Cp*TRIC). Overall, ion bindings associate with conformational changes within the TM segments of the trimeric channel, as evidenced from the comparisons of structures from various conditions (Supplementary Fig. 6). The mechanism for coupling ion binding to channel activity needs further investigation.

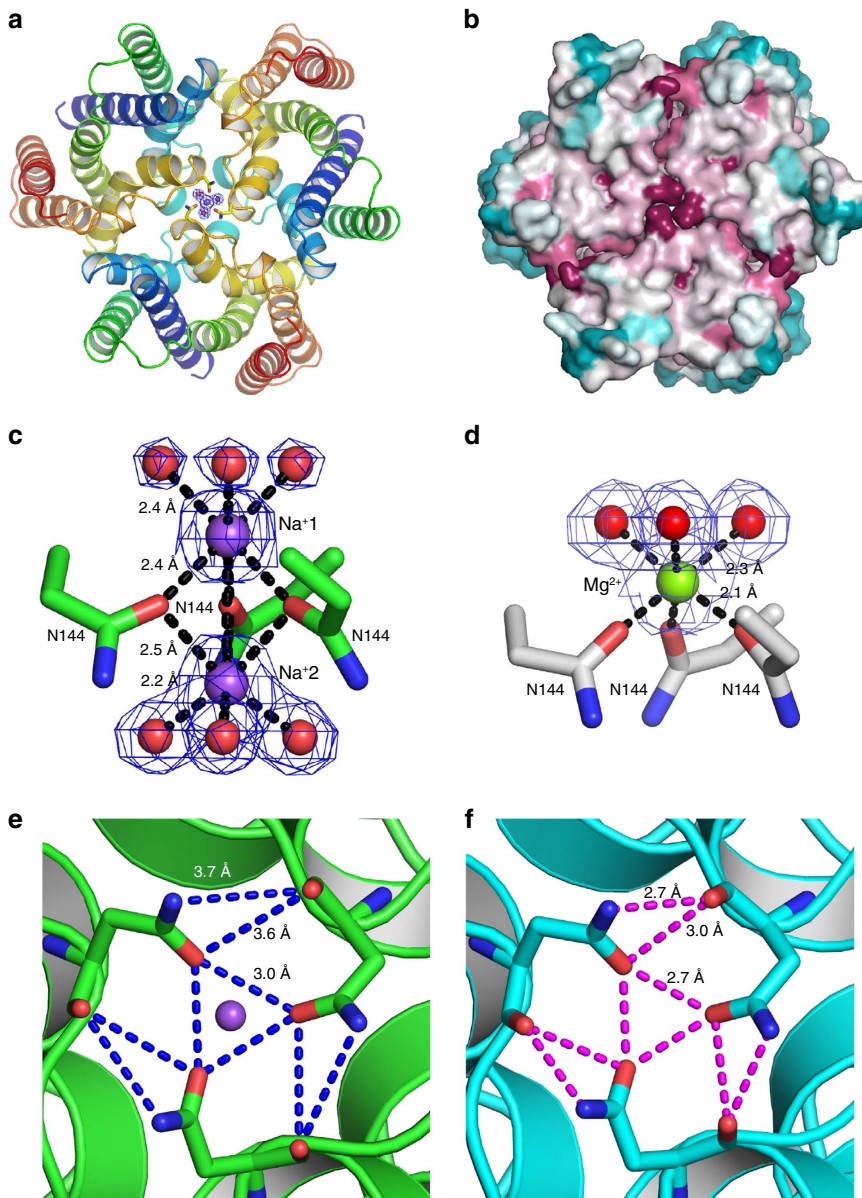

**Figure 7 | Ion binding and modulation of prokaryotic TRIC.** (**a**) Ribbon diagram of the SaTRIC trimer, coloured as in Fig. 1a but viewed from the cytosolic side. The conserved N144 residues along the three-fold axis of SaTRIC trimer (type 2a) are shown as stick, and bound Na$^+$ ions (purple) and water molecules (red) are shown as spheres. (**b**) Corresponding molecular surface for **a** coloured by sequence conservation as in Fig. 1b. (**c**) A zoom view of the N144, Na$^+$ ions and water molecules in the structure of SaTRIC (type 2a). Water molecules are shown as red spheres and bound Na$^+$ ions are shown as purple spheres. Asn144 $O_{\delta1}$ to Na$^+$1, distance = 2.4 Å; Na$^+$1 to water molecules, distance = 2.4 Å; Asn144 $O_{\delta1}$ to Na$^+$2, distance = 2.5 Å; and Na$^+$2 to water molecules, distance = 2.2 Å. Density contours are shown for both water molecules and Na$^+$ ions. (**d**) A zoom view of the N144, Mg$^{2+}$ ions and water molecules in the structure of SaTRIC (type 2b). Water molecules as in **c**, and Mg$^{2+}$ is shown as green sphere. Asn144 $O_{\delta1}$ to Mg$^{2+}$, distance = 2.1 Å; and Mg$^{2+}$ to water molecules, distance = 2.3 Å. Density contours are shown for both water molecules and Mg$^{2+}$ ion. (**e**) Close-up of bottom view of the N144 in the Na$^+$ bound structure of SaTRIC (type 2a). Bond distances are indicated. (**f**) Close-up bottom view of the N144 in the ion-free structure of SaTRIC (type 3). Bond distances are indicated.

**Distinctiveness of the TRIC channel structure.** The TRIC cation channels are structurally distinct and also appear to be gated in distinctively. The most remarkable feature for TRIC family architecture is that two inverted repeat motifs within each protomer help to form the ion conduction pore across the membrane. Interestingly, a similar six-helix topology (ignoring TM$_7$) and inverted repeat characteristics are also found in other homo-oligomeric channels, namely aquaporin (tetrameric) and formate channels (pentameric) (Supplementary Fig. 7a). The common feature for these inverted repeat motifs is three TMs, which are essentially straight for the first and third, whereas the kinked second TM of TRIC channels has the counterparts in aquaporin and formate channels that are broken into two halves, connected by an Ω loop (Supplementary Fig. 7b). As a consequence, although the topologies are similar, the structural overlaps of SaTRIC with AQP1 and FocA are highly divergent. If there is any evolutionary relationship among these inverted repeat structures of differing topologies, it is lost in the sequence record.

**Comparison to other TRIC channel structures.** Recently, structures were reported for two invertebrate TRIC channels[44]

and for two other prokaryotic TRIC channels[45]. Below, we compare those results to the structures that we report here.

*Caenorhabditis elegans* TRIC proteins, called TRIC-B1 and TRIC-B2 (Protein Databank (PDB) accession codes 5EGI and 5EIK), were analysed at resolutions of 3.3 and 2.3 Å, respectively[44], and are very similar to one another (r.m.s.d. of 1.1 Å for 212 superimposed $C_\alpha$ atoms). Both the protein foldings and trimeric organizations are also essentially the same as in our prokaryotic TRIC proteins. There are appreciable differences in details, however. Whereas *Sa*TRIC (type 2) and *Cp*TRIC are also very similar to one another (1.39 Å r.m.s.d. for 188 superimposed $C_\alpha$ atoms), the prokaryotic and invertebrate homologues differ considerably. For example, comparing *Ce*TRIC-B2 with *Sa*TRIC we find r.m.s.d. values of 3.33 (163 superimposed $C_\alpha$) and 2.48 Å (170 superimposed $C_\alpha$), respectively, for the type 2 and type 3 conformational states. Type 2 and type 3 *Sa*TRIC compare at 1.92 Å r.m.s.d. (188 superimposed $C_\alpha$). Although *Ce*TRIC-B2 is more similar to the open state of *Sa*TRIC, its pore is blocked.

The prokaryotic and invertebrate TRIC channels show conductance properties that are generally similar to one another, including sub-conductance states like those that are well characterized for vertebrate TRIC-B channels[33]. Other features differentiate the prokaryotic and invertebrate TRIC channels. In contrast to the endogenous $PIP_2$ molecules that are associated with the *Ce*TRIC proteins, we found no clear density for lipid molecules bound to the prokaryotic TRICs, although we did discover lipid-shaped fenestrations in *Sa*TRIC similar to those of *Rs*TRIC[45]. Also, the prokaryotic TRICs persist as trimers despite lacking tightly bound lipids, whereas the lack of $PIP_2$ destabilized the trimeric assembly of *Ce*TRIC-B1; and we found no electrophysiological evidence for 'monomers'. Ion-binding characteristics are also distinctive for these two classes of TRICs; in particular, there seems to be no cation site on the three-fold axis of *Ce*TRICs as in *Sa*TRIC, and $Ca^{2+}$-binding sites were not found for our prokaryotic TRICs. Finally, the hydrogen-bonded network of the locked C-THB end of the closed conformation of *Sa*TRIC structure is missing in the *Ce*TRICs. Almost nothing is known about the physiological roles of TRIC channels in either prokaryotes or invertebrates; however, it appears that although conductance properties are similar, their modes of regulation are likely different.

The additional prokaryotic TRIC structures are of bacterial *Rs*TRIC from *Rhodobacter sphaeroides* (3.4 Å resolution, PDB accession code 5H36) and archaeal *Ss*TRIC from *Sulfolobus solfataricus* (2.6 Å resolution, PDB accession code 5H35)[45]. These structures are very similar to one another and also to those of the prokaryotic TRICs reported here, with r.m.s.d. values of 1.24 Å for *Sa*TRIC (type 2a) versus *Ss*TRIC (184 superimposed $C_\alpha$) and 2.13 Å for *Sa*TRIC (type 3) versus *Ss*TRIC (183 superimposed $C_\alpha$). In keeping with the closer similarity of *Ss*TRIC to the closed conformation of *Sa*TRIC, an H-bonded network of interactions closes the C-THB in a locked conformation in both *Ss*TRIC and *Rs*TRIC, just as in the structures of both *Sa*TRIC type 2b and *Cp*TRIC. In addition, ion binding at the three-fold axis of the channel is also observed in *Ss*TRIC structure (N142), obtained in the presence of 100 mM $MgCl_2$ and 100 mM NaCl. The binding position, geometry and distances to side-chain atoms of *Ss*TRIC N142 are similar to those of the 1.9 Å-resolution structure of the $Mg^{2+}$-bound *Sa*TRIC (type 2b). These observations indicate conserved mechanisms for gating and modulation of prokaryotic TRICs.

## Methods

**Selection of target sequences.** *E. coli* YadS (now identified as *Ec*TRIC) was selected as a target for structural analysis by the NYCOMPS project in structural genomics of membrane proteins. This *Ec*TRIC sequence was used as a seed for expansion into other candidate family members by using standard NYCOMPS procedures[36]. PSI-BLAST[46] searches were made of the NYCOMPS98 data set of prokaryotic genomes, which contained ∼40,000 sequences for alpha-helical membrane proteins. Those sequences that matched *Ec*TRIC at $E < 10^{-3}$ in alignments with coverage of at least 50% of the predicted TM regions for both proteins were selected for further testing against our post-seed-expansion criteria[36]. Sequences meeting the criteria were submitted to the NYCOMPS protein production pipeline.

**Protein expression screening.** A total of 54 full-length TRIC homologues were identified and amplified by PCR from the following 42 prokaryotic species: *E. coli* K12; *Mesorhizobium loti*; *Bradyrhizobium japonicum* USDA110; *Xanthomonas campestris* ATCC 33913; *Vibrio fischeri* ES114; *Chromobacterium violaceum* DSM 30191; *Thermus thermophilus* HB27; *Silicibacter pomeroyi* DSS-3; *Salinibacter ruber* DSM 13855; *S. solfataricus*; *Streptomyces coelicolor* A3; *Rhodospirillum rubrum*; *Ralstonia solanacearum* GMI1000; *Pseudomonas fluorescens* Pf-5; *Pseudomonas syringae* DC3000; *Pseudomonas putida* KT2440; *Porphyromonas gingivalis* W83; *Pseudomonas aeruginosa* PAO1; *Desulfovibrio vulgaris* Hildenborough; *Deinococcus radiodurans* R1; *Corynebacterium glutamicum*; *C. psychrerythraea* 34H; *Agrobacterium tumefaciens* C58; *Bacteroides fragilis* NCTC 9343; *Acinetobacter sp.* ADP1; *Halobacterium sp.* NRC-1; *Haloarcula marismortui* DSM 3752; *Vibrio cholerae* N16961; *Bordetella bronchiseptica* RB50; *S. acidocaldarius*; *Lactococcus lactis*; *Streptococcus pyogenes* M1 GAS; *Streptococcus agalactiae* 2603V/R; *Salmonella typhimurium* LT2; *Erwinia carotovora atroseptica* SCRI1043; *Bacillus cereus* DSM 31; *Vibrio parahaemolyticus* RIMD 2210633; *Methanococcus maripaludis* S2; *Shigella flexneri* 2a 2457T; *Shewanella oneidensis* MR-1; *Bacillus subtilis* 168; and *Haemophilus influenzae* Rd KW20.

Selected cDNAs were cloned into a pET vector (Novagen, Inc.) that had been modified to fuse FLAG and deca-histidine sequences at the C terminus after a TEV protease cleavage site. Proteins were expressed in *E. coli* BL21(DE3) pLysS using a high-throughput format (0.6 ml volumes), and purified by metal-affinity capture after sonication in a buffer that contained *n*-dodecyl-β-D-maltopyranoside (Anatrace, Inc.). Analytical size-exclusion chromatography was then performed in 12 different detergent-containing mobile phases, among which were those with detergents *n*-dodecyl-β-D-maltopyranoside (DDM), *n*-decyl-β-D-maltopyranoside (DM), *n*-octyl-β-D-maltopyranoside (OM), *n*-octyl-β-D-glucopyranoside (OG) and lauryl dimethyl amine oxide. Multi-angle light scattering with refractive index detection was used to analyse the oligomeric state. The *Sa*TRIC (from *S. acidocaldarius*) and *Cp*TRIC (from *C. psychrerythraea*) were found to be monodisperse and stable, and these were sent on to be scaled up for structure determination.

**Scaled-up production and purification.** Scaled-up production of purified proteins adopted our previous procedures[47]. Transformed BL21 pLysS cells were grown to optical density (OD) 0.6–0.8 at 37 °C in 2× YT media after being inoculated with 1% of an overnight culture. After induction with 0.4 mM isopropyl-β-D-thiogalactoside, the culture was allowed to grow for another 4 h at 37 °C. Cells were then collected by centrifugation and stored at −80 °C for future use. Se-Met *Sa*TRIC mutant (L148M) and Se-Met *Cp*TRIC were expressed in a similar manner, but with Se-Met replacing methionine in defined minimal media. Cells were thawed, resuspended in a solubilization buffer (50 mM Tris-HCl, pH 8.0, and 200 mM NaCl), and then lysed with two passes in a French Press at 15–20,000 psi. Centrifugation at 10,000g for 20 min pelleted cell debris, and ultracentrifugation of the resulting supernatant at 150,000g for 1 h isolated the membrane fraction.

The membrane fraction was homogenized in the solubilization buffer and incubated at 4 °C for 1 h in DDM at a final concentration of 1% DDM (w/v). Ultracentrifugation at 150,000g for 30 min was used to remove non-dissolved matter. The supernatant was then loaded onto a 5 ml HiTrap $Ni^{2+}$-NTA affinity column (GE Healthcare), which had been pre-equilibrated with the solubilization buffer supplemented with 0.05% DDM. After a wash with 20 column volumes of solubilization buffer, His-tagged protein was eluted with 350 mM imidazole in the buffer. Affinity tags were removed by incubation at 4 °C overnight with TEV at 1:1,000 mass ratio, and tag removal was confirmed by SDS–PAGE. Samples were concentrated to ∼10 mg ml⁻¹, and preparative size-exclusion chromatography on a Superdex-200 column was performed to purify detergent-solubilized protein from aggregates, TEV protease and cleaved tags, and for buffer and detergent exchange. The gel-filtration buffer contained 20 mM HEPES (pH 7.5), 200 mM NaCl (or RbCl or NaBr; Supplementary Table 2) and 2× CMC (critical micelle concentration) of detergent. The *Sa*TRIC and *Cp*TRIC proteins were well behaved and stable in nearly all tested detergents, and we have purified them from DDM, DM, OM, OG and lauryl dimethyl amine oxide.

**Biochemical characterization.** For crosslinking experiments, purified *Sa*TRIC and *Cp*TRIC protein in DDM (around 1 mg ml⁻¹ in 30 μl) was incubated with various concentrations of glutaraldehyde (0, 0.00005%, 0.0001%, 0.0005% and 0.001%) at room temperature for 8 h. The reaction was quenched by addition of 50 mM Tris-Cl, pH 8.0. The incubated samples were separated on 12% SDS–PAGE gels, showed a ladder consistent with a trimeric structure.

**Crystallization and data collection.** Purified *Sa*TRIC and *Cp*TRIC protein in a series of detergents, including DDM, DM, OM and OG, were concentrated to ∼10 mg ml$^{-1}$ for initial crystal trials in a Mosquito robot with commercial screens from Hampton research, Emerald Biosystems and Molecular Dimension. *Sa*TRIC protein readily crystallized in detergent micelles, but diffracted poorly. Improved diffraction was obtained after the application of the LCP crystallization. The *Sa*TRIC, purified in detergent DM, was reconstituted into the LCP by mixing with monoolein at a 2:3 protein to lipid ratio (w/w), using the twin-syringe mixing method. After the reconstitution, 50 nl LCP drops were dispensed onto laminex glass plates and manually overlaid with 1.0 μl of precipitant solution. After extensive optimization we reached conditions supporting very high resolution. Wild-type *Sa*TRIC crystallized from 100 mM Li$_2$SO$_4$, 50 mM Na-Citrate (pH 5.5) and 40% PEG200. To facilitate phase determination by Se-SAD, we introduced one methionine mutation (L148M) in the wild-type *Sa*TRIC. The Se-Met *Sa*TRIC (L148M) was also crystallized in LCP, but in a different condition that contain the following: 40% PEG200; 0.1 M NaCl; 0.1 M MgCl$_2$; and 0.1 M MES (pH 6.0). Cryo-protection was achieved by adding 5% PEG200 to the crystallization solution. The Se-Met *Cp*TRIC, when solubilized in β-octylmaltoside (OM), was crystallized in detergent micelles in the solution containing 38% PEG400, 0.1 M NaCl, 0.1 M CdCl$_2$ and 0.1 M Tris-HCl (pH 8.5). For other crystals of *Sa*TRIC and *Cp*TRIC, see Supplementary Table 2 for details.

**Structure determination and refinement.** Native and Se-Met SAD data were collected at multiple beamlines, including BL17U and BL19U at Shanghai Synchrotron Radiation Facility and at 24IDC at Advanced Photon Source and processed with the software XDS[48]. Crystals of the Se-Met *Sa*TRIC (L148M) mutant were grown in space group of P321, with cell dimensions $a = b = 111.1$ Å, $c = 47.7$ Å and $α = β = 90°$, $γ = 120°$. The initial phase was determined at 3.1 Å resolution from Se-Met SAD with data collected at 0.97853 Å. Assessment of data quality for phasing, location of heavy atom sites and phases were calculated using the HKL2MAP interface to SHELX programmes. All the secondary structure elements were clearly visible in the experimental electron density map. The initial model was built manually using COOT[49]. Native crystals for the wild-type *Sa*TRIC protein belong to space group of P6$_3$ with cell dimensions $a = b = 64.3$ Å, $c = 80.8$ Å and $α = β = 90°$, $γ = 120°$. Phases for the native wild-type *Sa*TRIC structure (type 2a) were determined by molecular replacement using the model obtained by Se-Met SAD data and extended to 1.60 Å resolution. The native model was rebuilt in COOT according to 2$F_O F_C$ and 1$F_O F_C$ map and refined at 1.60 Å resolution against the wild-type *Sa*TRIC native data with phenix.refine[50]. Structure validation was performed with PROCHECK[51]. Data collection and refinement statistics are summarized in Table 1, and figures were prepared in PyMOL (http://www.pymol.org).

Other structures, including Mg$^{2+}$-bound *Sa*TRIC (type 2b), the ion-free *Sa*TRIC (type 3) and other ion complexes were solved by molecular replacement by using crystals obtained from LCP under different conditions; see Supplementary Table 2 for details. Identifications of the different ions in these structures were based on the coordination geometries and on refinements of B-factors and occupancies for alternative elements corresponding to ions that were present in the crystallization media. In the *Sa*TRIC structure from the type 2a crystal, the two Na$^+$ ions along the three-fold axis of the trimeric channel were at the distance of 2.4–2.5 Å to the neighbouring atoms of surrounded residues or waters, and the refined B-factors were 22.8 for Na$^+$1 and 27.2 for Na$^+$2, which were similar to that of the neighbouring atoms; however, if we replaced the Na$^+$ ion with another possible ion species in the solution, Li$^+$, the refined B-factor became 6.1 and 6.4 respectively, which was far smaller than that of the neighbouring atoms. In another *Sa*TRIC structure from the type 2b crystal, the assigned Mg$^{2+}$ ion along the three-fold axis of the trimeric channel was at the distance of 2.1–2.3 Å to the neighbouring atoms of surrounded residues or waters, with a refined B-factor of 19.3, which was similar to that of the neighbouring atoms. If we replaced the Mg$^{2+}$ ion with another possible ion species in the solution, Rb$^+$ ion, the refined B-factor became 68.6, which was far larger than that of the neighbouring atoms.

*Cp*TRIC crystals have a space group of R32, with cell dimensions $a = b = 91.3$ Å and $c = 252.8$ Å, one protein molecule per ASU. The structure was solved at 2.4 Å by Se-Met SAD. The resulting electron density map permitted automatic tracing of a nearly complete model.

**Site-directed mutagenesis.** A QuikChange site-directed mutagenesis kit (Stratagene, La Jolla, CA) was used to construct mutants, which were confirmed by sequencing. Mutants were expressed and purified as for the wild-type protein.

**Electrophysiology.** Lipid bilayer experiments were conducted as in our study of a bacterial bestrophin[52]. *Sa*TRIC proteins, including wild type and mutants, were purified in the same way as those used for crystallization. Purified *Sa*TRIC proteins (wild type and mutants), at final concentration between 1 and 5 μg ml$^{-1}$, were fused into planar lipid bilayers formed by painting a lipid mixture of phosphatidylethanolamine and phosphatidylcholine (Avanti Polar Lipids) in a 3:1 ratio in decane across a 200 μm hole in a polystyrene partition separating the internal and external solutions in polysulfonate cups (Warner Instruments). The *trans* chamber (1.0 ml), representing the extracellular compartment, was connected

to the head stage input of a bilayer voltage clamp amplifier. The *cis* chamber (1.0 ml), representing the cytoplasmic compartment, was held at virtual ground. Solutions used for I–V or cation/anion channel detection were as follows (in mM): 210 KCl and 10 HEPES (pH 7.4) in the *cis* solution; and 210/810 KCl and 10 HEPES (pH 7.4) in the *cis/trans* solution. For ion selectivity experiments, the *trans*-side KCl was changed to either 210 mM XCl (X = Na$^+$, or Rb$^+$, Cs$^+$) or 105 mM YCl$_2$ (Y = Ca$^{2+}$ or Mg$^{2+}$). Purified proteins were added to the *cis* side and fused with the lipid bilayer. Currents were recorded every 2 min after application of the voltage to the *trans* side. Single-channel currents were recorded using a Bilayer Clamp BC-525D (Warner Instruments, LLC, CT), filtered at 1 kHz using a Low-Pass Bessel Filter 8 Pole (Warner Instruments, LLC, CT) and digitized at 4 kHz. All experiments were performed at room temperature (23 ± 2 °C). pH for the solution was adjusted with Choline. The membrane potential that gives zero current during ramp pulse was taken as the reversal potential

**Electrophysiological data and statistical analyses.** The relative monovalent cation permeability (PX/PK) was calculated according to the Goldman–Hodgkin–Katz equation:

$$PX^+/PK^+ = [K^+]/[X^+] \exp(-\Delta E_{rev}F/RT). \tag{1}$$

The relative divalent cations (Ca$^{2+}$ and Mg$^{2+}$) versus K$^+$ permeability ratio (PX$^{2+}$/PK$^+$) of *Sa*TRIC was calculated using the Fatt–Ginsborg equation[53]:

$$PX^{2+}/PK^+ = [K^+]/4[X^{2+}] \exp(-\Delta E_{rev}F/RT) [\exp(-\Delta E_{rev}F/RT) + 1] \tag{2}$$

where R, T and F have their usual meanings and $E_{rev}$ is the zero current reversal potential. The value of RT/F used was 25.7 mV at 23 °C.

Single-channel Po was determined over 2 min of continuous recording using the method of 50% threshold analysis. The recordings were analysed by using Clampfit 10.1 (Molecular Devices) and Prism (ver.5.0, GraphPad).

**Data availability.** The X-ray crystallographic coordinates and structure factor files of the prokaryotic TRIC structures have been deposited in the PDB with the accession codes 5WUC, 5WUD, 5WUE and 5WUF. The following PDB accession codes were used in this work: 5EGI (*Ce*TRIC-B1); 5EIK (*Ce*TRIC-B2); 5H36 (*Rs*TRIC); and 5H35 (*Ss*TRIC). All other data supporting the finding of this study are either provided in the Article and Supplementary Files or available from the authors on reasonable request.

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

## Acknowledgements

We thank Bailong Xiao and Jian Yang for helpful discussions; James Love and staff of the New York Consortium on Membrane Protein Structure (NYCOMPS) for assistance in early stages of the project; laboratory colleagues for advice in protein chemistry and crystallography; and Wenming Qin, Deqiang Yao, Lijie Wu, Oliver Clarke, Shuaiyi Liang and Wei Wang for help with synchrotron experiments at beamlines, BL19U and BL17U at Shanghai Synchrotron Radiation Facility, or NE-CAT beam-line 24-ID-C at the Advanced Photon Source. This project is financially supported by the Strategic Priority Research Program of the Chinese Academy of Sciences (XDB08020301), the National 973 Project Grant of the Ministry of Science and Technology (2015CB910102 and 2016YFA0500503) and the National Natural Science Foundation of China (31322005) to Y.C. Y.C. is supported in part by the 'National Thousand Young Talents' programme from the Office of Global Experts Recruitment in China. W.A.H. is supported in part by NIH grant GM 107462. This work also benefited from NIH support to the New York Consortium on Membrane Protein Structure (NYCOMPS; U54 GM095315) and by the Center on Membrane Protein Production and analysis (COMPPÅ; P41 GM116799).

## Author contributions

M.S. performed bioinformatics analyses, purified the *Sa*TRIC proteins, performed LCP crystallization, collected data at Advanced Photon Source beamlines, conducted bilayer measurements and analysed the structure; Y.M. purified *Sa*TRIC and *Cp*TRIC proteins, performed crystallizations in detergent micelles and collected data at Shanghai Beamlines; Q.Y. conducted bilayer measurements and analysed the data; F.G. collected diffraction data, solved and analysed the structure; D.-l.L. purified *Sa*TRIC wild-type and mutant proteins and collected data at Shanghai Beamlines; Y.G., X.-h.W., R.B., B.K., C.Y., H.Z., Y.Z. and F.-b.Z. performed experiments; A.R.M. oversaw analyses of bilayer measurements; W.A.H. analysed structure and wrote the manuscript; Y.-h.C. initiated the project, planned and analysed experiments, supervised the research and wrote the manuscript. All authors contributed to discussion of the data and editing of the manuscript.

## Additional information

**Competing interests:** The authors declare no competing financial interests.

