## [Peer Review File · Nature Communications]

Reviewer #1 (Remarks to the Author)

This manuscript describes the structures and electrophysiological characterization of prokaryotic homologs of TRIC cation channels. These channels play key roles in muscle excitation-contraction coupling and other processes, by providing countercurrents with respect to calcium release channels such as Ryanodine Receptors.

The authors show structures of two prokaryotic TRIC channels, which both form trimeric arrangements. Within each trimer, there are three pores, and each subunit is built up by 7 TM helices. Different conformations for the SaTRIC channel are crystallized, interpreted as being open and closed. Opening transitions seem to occur by virtue of a tri-glycine repeat in the middle of two helices, resulting in altered kinking. Co-crystallization with different ions unveil a few different ion binding sites. Planar lipid bilayer experiments investigate ion selectivity, and show the importance of a number of residues in modulating the single channel conductance and open probability. A distinct set of substates are also observed.

Assessment:

1) The authors present the first prokaryotic TRIC structures, but unfortunately not the first TRIC structures, since two TRIC channels from *C.elegans* were published last month in Nature. This takes away some of the novelty, however I do think the current paper has several new things to add, thanks to a higher resolution and different gating transitions from those described for *C.elegans*.

Although the *C.elegans* structures got published while the current manuscript was under preparation, I suggest the authors to not ignore it, but to include an extra section on comparing the structures and gating transitions.

2) I have a major issue with the electrophysiology data. In particular, the text and figure 5 legend mention that an 'opposite current' is obtained for D99A relative to wild-type, and that this is indicative of a switch to chloride ion conductance instead of potassium. This does not make any sense. The legend for figure 5f clearly states that a symmetrical gradient of KCl is used (210mM in both cis and trans), with the application of a +20mV potential difference. Under these conditions, a switch in selectivity from K⁺ to Cl⁻ would not reveal any different current. K⁺ would flow towards the negative potential, whereas Cl⁻ would flow toward the positive. Electrically, there is no difference between positive charges flowing in one direction and negative charges in the opposite one. The authors thus made a fundamental mistake, and unfortunately this does make me question the validity and interpretation of the other electrophysiology data.

3) Several structures are shown with different ions bound. There is minimal to no explanation as to how particular densities were assigned to these. In most cases, there are different cations present in the crystallization mix. Ideally, some anomalous diffraction experiments would be useful, but minimally a thorough explanation in the methods would be good, e.g. a complete description of the coordination geometry, and of the expected coordination geometry of each ion that is present in the crystallization mix. Some distances are already mentioned, but this should be more comprehensive.

4) Figure 6b shows a cross-section of an 'open state' TRIC. However, the opening does not seem continuous in this image. So is this absolutely guaranteed to be an open channel? If the figure doesn't capture something that's readily visible in 3D, it would be good to see this. However, if the channel is truly not continuous, it may suggest that a gate in the middle is still closed, and this possibility should be properly discussed.

Minor comments:

1) Page 6, last line: 'DM' is used for dodecylmaltoside. Either this should be DDM, or it should be decylmaltoside. Please correct and also fact-check the corresponding methods section.

2) page 5: "we clustered nearly 425..". Just state the exact number.

3) It would be useful to have some comparisons with the mammalian TRICs at various locations. E.g. the electrostatic surface mentioned on page 8: would this be present in humans? Page 10 mentions a hydrogen bond network in C-THB that is involved in cytoplasmic closure. From the sequence alignment that's presented, this network doesn't seem to exist in the human counterparts. It would be good to see this discussed explicitly (and also compared with the C.elegans TRIC structures).

4) The supplementary note 1 is very confusing and could use a significant expansion.

5) Table 1: The number of Ramachandran outliers for the CpTRIC structures are not mentioned.

Reviewer #2 (Remarks to the Author)

Su et al. describe crystallization studies of SaTRIC and CpTRIC, focusing on the structure of SaTRIC. The authors provide accompanying functional data where purified SaTRIC preparations were reconstituted into bilayers using recording conditions similar to those used by other groups describing the properties of mammalian TRIC channels. The study provides an interesting comparison to the recently published structure of the C.elegans TRIC-B1 and TRIC-B2 by Yang et al. (2016).

I have the following comments:

1. There should be more comparisons made between the structure and function of the TRIC proteins described in this manuscript and those of Yang et al. (2016). For example, there is PIP2 associated with C. elegans TRIC but not with SaTRIC or CpTRIC. The functional, bilayer data does not look similar to the C. elegans TRIC and this should be discussed.

1. Figure 2 is difficult to understand because the text and figure legend switches between the terms 'extracellular' and 'luminal', 'inside' and 'outside' etc. These are SR proteins so it would be more appropriate if the terminology was consistent throughout the whole manuscript describing the orientation of the TRIC molecules as the cytoplasmic and luminal sides.

2. The methods do not explicitly describe whether the purified proteins used in bilayer experiments are prepared in the same way as the proteins used for crystallization. This is important if correlations between structure and function are to be made. The methods for ion channel reconstitution need to be explained in full detail.

3. There are no controls for functional data. How do the authors rule out the possibility that the single channel events are not due to contaminants or to detergents?

4. page 12. With regard to the permeability of TRIC-B (line 1) and multiple conductance levels for TRIC-B (lines 7 and 8) the wrong papers are referenced. In line 1, reference 23 should be replaced by Venturi et al. 2013. In lines 7/8 all the references are incorrect and should be replaced by Venturi et al. 2013 and Matyjaszkiewicz et al. 2015, Biophys. J. 109, 265-76. The same is true for page 11, line 19. Reference 23 should be replaced by Venturi et al. 2013.

5. (i) Figure 5a shows representative traces of individual TRIC channel traces which is fine but it looks like Figure 5b is also a representative experiment. Figure 5b should show results for multiple experiments and include error bars. In gradient, there should be measurements at negative voltages to demonstrate the reversal potential clearly in case there is any asymmetry.

(ii) the experimental details for Figure 5c should be fully described; what solutions were used and which side of the bilayer were they present on? The current-voltage plots (with error bars) showing reversal potentials should be shown for each combination of solutions in the Supplementary material.

The wrong equation was used to calculate relative permeability ratios for divalent versus monovalent cations. The Fatt & Ginsborg equation should be used. The Yang et al. (2016) paper

finds that millimolar Ca^{2+} opens or closes TRIC depending which side of the channel is exposed to it. Presumably millimolar Ca^{2+} was present on one or other side of the channel to perform the relative permeability experiments. What effect does millimolar Ca^{2+} have on the activity of SaTRIC?

(iii) Figures 5g and h. The legends are muddled. Figure 5g shows conductance not P_o and visa versa. In fact, the y axis should not read 'conductance' but 'current amplitude at +20 mV'. The number of replicates for each bar in both bar charts should be placed above each bar.

(iv) If the authors wish to include statements (page 12, lines 15 -19) about the mutant D99A passing Cl^- current, which is very interesting, then solid experiments (full current-voltage plots showing reversal potentials in gradients) with replicates to prove this should be provided.

Otherwise how do the authors know that they are not just incorporating junk into the bilayer? The bar chart in Fig 5g which reads 'conductance' but should read 'current amplitude at +20 mV' should reflect that current is not in the same direction as the other bars.

(v) The section entitled "alternative trimer sub-conductance models' on the last page of the supplementary information is speculative and poorly described and should be omitted.

Response to Referees

Referees:

We copy the referee comments below verbatim and respond to each point, describing how we have modified the paper to address the concern. We identify the locations of major changes below and we highlight all changes in the text file.

Reviewer #1 (Remarks to the Author):

This manuscript describes the structures and electrophysiological characterization of prokaryotic homologs of TRIC cation channels. These channels play key roles in muscle excitation-contraction coupling and other processes, by providing countercurrents with respect to calcium release channels such as Ryanodine Receptors.

The authors show structures of two prokaryotic TRIC channels, which both form trimeric arrangements. Within each trimer, there are three pores, and each subunit is built up by 7 TM helices. Different conformations for the SaTRIC channel are crystallized, interpreted as being open and closed. Opening transitions seem to occur by virtue of a tri-glycine repeat in the middle of two helices, resulting in altered kinking. Co-crystallization with different ions unveil a few different ion binding sites. Planar lipid bilayer experiments investigate ion selectivity, and show the importance of a number of residues in modulating the single channel conductance and open probability. A distinct set of substates are also observed.

Assessment:

1) The authors present the first prokaryotic TRIC structures, but unfortunately not the first TRIC structures, since two TRIC channels from *C.elegans* were published last month in Nature. This takes away some of the novelty, however I do think the current paper has several new things to add, thanks to a higher resolution and different gating transitions from those described for *C.elegans*.

Although the *C.elegans* structures got published while the current manuscript was under preparation, I suggest the authors to not ignore it, but to include an extra section on comparing the structures and gating transitions.

Response: We have added an extra section in the Discussion to compare our prokaryotic TRIC structures and functional results with those published for TRIC channels from *C. elegans* (pp.16-18).

2) I have a major issue with the electrophysiology data. In particular, the text and figure 5 legend mention that an 'opposite current' is obtained for D99A relative to wild-type, and that this is indicative of a switch to chloride ion conductance instead of potassium. This does not make any sense. The legend for figure 5f clearly states that a symmetrical gradient of KCl is used (210mM in both cis and trans), with the application of a +20mV potential difference. Under these conditions, a switch in selectivity from K⁺ to Cl⁻ would not reveal any different current. K⁺ would flow towards the negative potential, whereas Cl⁻ would flow toward the positive. Electrically, there is no difference between positive charges flowing in one direction and negative charges in the opposite one. The authors thus made a fundamental mistake, and unfortunately this does make me question the validity and interpretation of the other electrophysiology data.

Response: Given what we wrote, the referee raises a completely valid concern. Unfortunately, the legend that we presented for former Fig. 5f was mistaken. In fact, this figure presents data from an asymmetric gradient across the membrane (210mM KCl cis, 810mM KCl trans), and these data do support potassium conductance by the wild-type protein and a switch to chloride conductance for D99A. We apologize for the confusing mixup. We have revised Figure 5, again showing these results from the asymmetric gradient (new Fig. 5f) and now also showing results from a symmetrical gradient (210mM KCl in both chambers) at +20mV potential (new Fig. 5e). We have also revised the text and legend accordingly (p.12 and p. 34).

3) Several structures are shown with different ions bound. There is minimal to no explanation as to how particular densities were assigned to these. In most cases, there are different cations present in the crystallization mix. Ideally, some anomalous diffraction experiments would be useful, but minimally a thorough explanation in the methods would be good, e.g. a complete description of the coordination geometry, and of the expected coordination geometry of each ion that is present in the crystallization mix. Some distances are already mentioned, but this should be more comprehensive.

Response: The point is well taken that we were not sufficiently clear about the identification of ions found in these structures, and we accept that descriptions of our methods were not very clear. Unfortunately, we do not have appropriate anomalous diffraction data available in all cases (we do conclusively identify Cd²⁺ sites for CpTRIC, **Supplementary Fig. 5**); however, we do believe that identifications from refinements of B factors and occupancies for alternative ions that are present in the crystallization media are definitive in their own right. In addition, the coordinating ligands and coordination geometries are consistent with these ion identifications. We describe these procedures in the methods (p. 23).

4) Figure 6b shows a cross-section of an ‘open state’ TRIC. However, the opening does not seem continuous in this image. So is this absolutely guaranteed to be an open channel? If the figure doesn’t capture something that’s readily visible in 3D, it would be good to see this. However, if the channel is truly not continuous, it may suggest that a gate in the middle is still closed, and this possibility should be properly discussed.

Response: The referee raises a valid point of concern, which arises from confusions generated by old Figure 6. The channel is indeed open based on calculations by program *Hole*. Unfortunately, the previous rendering of the *Hole* surface was unsatisfactory and misleading. We have redrawn the figure to better capture the reality.

Minor comments:

1) Page 6, last line: ‘DM’ is used for dodecylmaltoside. Either this should be DDM, or it should be decylmaltoside. Please correct and also fact-check the corresponding methods section.

Response: The text should have read “decylmaltoside (DM),” as we have it now. Thank you for catching this mistake.

2) page 5: “we clustered nearly 425..”. Just state the exact number.

Response: We now have the exact number, which is 425 (p.5).

3) It would be useful to have some comparisons with the mammalian TRICs at various locations. E.g. the electrostatic surface mentioned on page 8: would this be present in humans? Page 10 mentions a hydrogen bond network in C-THB that is involved in cytoplasmic closure. From the sequence alignment that’s presented, this network doesn’t seem to exist in the human counterparts. It would be good to see this discussed explicitly (and also compared with the *C.elegans* TRIC structures).

Response: These are good suggestions, and we have modified the descriptions to present distinctions and implications for mammalian TRICs (e.g. p. 10 and pp. 17-18 on the C-THB network). On some particulars, such as electrostatic potential surfaces, we could in principle generate a homology model from which this surface could be produced. We wish to refrain from this exercise here because we have recently solved structures of vertebrate TRIC channels, which essentially solve this problem. It would be beyond the scope of this paper to include those results here and it would be disingenuous and pointless to produce a homology model when we basically know the true answer.

4) The supplementary note 1 is very confusing and could use a significant expansion.

Response: We have eliminated this supplementary note as advocated by Referee #2.

5) Table 1: The number of Ramachandran outliers for the CpTRIC structures are not mentioned.

Response: We have added the missing Ramachandran outlier number, which is 0 for each of the CpTRIC structures. We have also decided to eliminate mention of the Rb⁺ data set since we found no evidence for bound Rb⁺ and had no descriptions of this structure, which is isomorphous with the SeMet structure, which we do describe.

Reviewer #2 (Remarks to the Author):

Su et al. describe crystallization studies of SaTRIC and CpTRIC, focusing on the structure of SaTRIC. The authors provide accompanying functional data where purified SaTRIC preparations were reconstituted into bilayers using recording conditions similar to those used by other groups describing the properties of mammalian TRIC channels. The study provides an interesting comparison to the recently published structure of the *C.elegans* TRIC-B1 and TRIC-B2 by Yang et al. (2016).

I have the following comments:

1. There should be more comparisons made between the structure and function of the TRIC proteins described in this manuscript and those of Yang et al. (2016). For example, there is PIP2 associated with *C. elegans* TRIC but not with SaTRIC or CpTRIC. The functional, bilayer data does not look similar to the *C. elegans* TRIC and this should be discussed.

Response: The paper from Yang *et al.* (2016) on TRIC proteins from *C. elegans* appeared online just a day before we submitted our paper. We agree that our revised manuscript should compare our results with those in this publication, and we now make such comparisons in a concluding section of our Discussion (pp. 16-18).

1. Figure 2 is difficult to understand because the text and figure legend switches between the terms ‘extracellular’ and ‘luminal’, ‘inside’ and ‘outside’ etc. These are SR proteins so it would be more appropriate if the terminology was consistent throughout the whole manuscript describing the orientation of the TRIC molecules as the cytoplasmic and luminal sides.

Response: We agree that it was confusing for us to have been inconsistent in the nomenclature regarding to membrane surfaces, both in Figure 2 and also in other parts of the submitted manuscript. We now describe these prokaryotic proteins only in relation to extracellular and cytoplasmic (or intracellular) sides, and simply relate these orientations to the anticipated orientations in the SR and ER membranes of mammalian TRIC channels.

2. The methods do not explicitly describe whether the purified proteins used in bilayer experiments are prepared in the same way as the proteins used for crystallization. This is important if correlations between structure and function are to be made. The methods for ion channel reconstitution need to be explained in full detail.

Response: We apologize for being unclear about the protein preparations and reconstitution methods. Yes, the protein samples used in lipid bilayer reconstitutions were prepared in the same way as those used for crystallization. We now make this clear in the Methods section where we also elaborate on the methods used for ion channel reconstitution (p. 24).

3. There are no controls for functional data. How do the authors rule out the possibility that the single channel events are not due to contaminants or to detergents?

Response: We did conduct control experiments with the buffers used for the purified proteins, including detergents. All electrophysiology experiments were performed on proteins captured by metal-affinity chromatography and shown to be clean of contaminants by size exclusion chromatography. The controls and these procedures are better described in the revised manuscript (p. 24). Moreover, the same purification and bilayer incorporation procedures were used for all samples but yet the conductance traces are distinctive and roughly similar to the properties of mammalian TRIC channels.

4. page 12. With regard to the permeability of TRIC-B (line 1) and multiple conductance levels for TRIC-B (lines 7 and 8) the wrong papers are referenced. In line 1, reference 23 should be replaced by Venturi et al. 2013. In lines 7/8 all the references are incorrect and should be replaced by Venturi et al. 2013 and Matyjaszkiewicz et al. 2015, *Biophys. J.* 109, 265-76. The same is true for page 11, line 19. Reference 23 should be replaced by Venturi et al. 2013.

Response: We apologize for citing the wrong papers and appreciate being corrected on this. We have carefully reviewed this section and believe that we have properly cited the prior literature in the revised paper (pp. 11-12).

5. (i) Figure 5a shows representative traces of individual TRIC channel traces which is fine but it looks like Figure 5b is also a representative experiment. Figure 5 b should show results for multiple experiments and include error bars. In gradient, there should be measurements at negative voltages to demonstrate the reversal potential clearly in case there is any asymmetry.

Response: In fact, old Fig. 5b was compiled from multiple experiments on wild-type SaTRIC. We agree that error bars should have been included and we do so now. We have also extended measurements into negative voltages for the gradient condition, including error bars here as well.

(ii) the experimental details for Figure 5c should be fully described; what solutions were used and which side of the bilayer were they present on? The current-voltage plots (with error bars) showing reversal potentials should be shown for each combination of solutions in the Supplementary material. The wrong equation was used to calculate relative permeability ratios for divalent versus monovalent cations. The Fatt & Ginsborg equation should be used. The Yang et al. (2016) paper finds that millimolar Ca^{2+} opens or closes TRIC depending which side of the channel is exposed to it. Presumably millimolar Ca^{2+} was present on one or other side of the channel to perform the relative permeability experiments. What effect does millimolar Ca^{2+} have on the activity of SaTRIC?

Response: We apologize for not providing adequate descriptions for the relative permeability experiments used to produce Fig. 5c. In the revised manuscript, we attempt to describe these experiments completely and we now show I-V plots supporting each of the ion permeabilities. We have also redone the analysis for divalent cations using the Fatt & Ginsborg equation. Thank you for correcting us. We did not study the effect of millimolar Ca^{2+} on the activity of SaTRIC; however, we did check for the binding of Ca^{2+} to SaTRIC by soaking Type 2a crystals in 10 mM and found no evidence for bound Ca^{2+} ions.

(iii) Figures 5g and h. The legends are muddled. Figure 5g shows conductance not P_o and visa versa. In fact, the y axis should not read 'conductance' but 'current amplitude at +20 mV'. The number of replicates for each bar in both bar charts should be placed above each bar.

Response: We apologize for interchanging the legends for Figures 5g and 5h, and we have corrected this mistake in the revised manuscript. There were four replicates in each case, and we indicate this by adding "n = 4" above the bars.

(iv) If the authors wish to include statements (page 12, lines 15 -19) about the mutant D99A passing Cl^- current, which is very interesting, then solid experiments (full current-voltage plots showing reversal potentials in gradients) with replicates to prove this should be provided. Otherwise how do the authors know that they are not just incorporating junk into the bilayer? The bar chart in Fig 5g which reads 'conductance' but should read 'current amplitude at +20 mV' should reflect that current is not in the same direction as the other bars.

Response: Although bilayer reconstitutions with the D99A protein led were short lived recordings, we did succeed in measuring full I-V data as suggested, which we present in Supplementary Figures 2e-f. We are confident that we "are not just incorporating junk into the bilayer" for reasons discussed above in response to concern 3; our controls show no currents and our purifications show no proteins other than TRIC. We changed the axial label for as suggested (now Fig. 5h).

(v) The section entitled "alternative trimer sub-conductance models' on the last page of the supplementary information is speculative and poorly described and should be omitted.

Response: We have removed this speculation from the revised manuscript.

Reviewer #1 (Remarks to the Author)

The authors have addressed all my previous concerns and comments. Most importantly, the electrophysiology description has improved, and comparisons have been added with recently reported TRIC structures. The manuscript should be acceptable for publication.

Reviewer #2 (Remarks to the Author)

There is still one mistake with the referencing on page 12, line 4: 'roughly similar to mouse TRIC-B'. Reference 32 (Pitt et al) is incorrect. References 33 and 43 should be used here.

Otherwise, this very interesting manuscript is much improved.

Point-by-Point Response to Final Reviewer Comments

Reviewer #1 (Remarks to the Author):

The authors have addressed all my previous concerns and comments. Most importantly, the electrophysiology description has improved, and comparisons have been added with recently reported TRIC structures. The manuscript should be acceptable for publication.

Response: We are grateful for the improvements that your comments have generated and pleased that you find them acceptable.

Reviewer #2 (Remarks to the Author):

There is still one mistake with the referencing on page 12, line 4: 'roughly similar to mouse TRIC-B'. Reference 32 (Pitt et al) is incorrect. References 33 and 43 should be used here.

Otherwise, this very interesting manuscript is much improved.

Response: We apologize for inadequate proofreading and are grateful for the correction. We did intend to cite Venturi *et al.* (Reference 33) here, and not Pitt *et al.* (Reference 32), and we have made the correction: "roughly similar to mouse TRIC-B^{33,43}."